# BOOSTING RANDOMIZED SMOOTHING WITH VARIANCE REDUCED CLASSIFIERS

**Miklós Z. Horváth, Mark Niklas Müller, Marc Fischer, Martin Vechev**
Department of Computer Science
ETH Zurich, Switzerland
`mihorvat@ethz.ch, {mark.mueller,marc.fischer,martin.vechev}@inf.ethz.ch`

## ABSTRACT

Randomized Smoothing (RS) is a promising method for obtaining robustness certificates by evaluating a base model under noise. In this work, we: (i) theoretically motivate why ensembles are a particularly suitable choice as base models for RS, and (ii) empirically confirm this choice, obtaining state-of-the-art results in multiple settings. The key insight of our work is that the reduced variance of ensembles over the perturbations introduced in RS leads to significantly more consistent classifications for a given input. This, in turn, leads to substantially increased certifiable radii for samples close to the decision boundary. Additionally, we introduce key optimizations which enable an up to 55-fold decrease in sample complexity of RS for predetermined radii, thus drastically reducing its computational overhead. Experimentally, we show that ensembles of only 3 to 10 classifiers consistently improve on their strongest constituting model with respect to their average certified radius (ACR) by 5% to 21% on both CIFAR10 and ImageNet, achieving a new state-of-the-art ACR of $0.86$ and $1.11$, respectively. We release all code and models required to reproduce our results at `https://github.com/eth-sri/smoothing-ensembles`.

## 1 INTRODUCTION

Modern deep neural networks are successfully applied to an ever-increasing range of applications. However, while they often achieve excellent accuracy on the data distribution they were trained on, they have been shown to be very sensitive to slightly perturbed inputs, called adversarial examples (Biggio et al., 2013; Szegedy et al., 2014). This limits their applicability to safety-critical domains. Heuristic defenses against this vulnerability have been shown to be breakable (Carlini & Wagner, 2017; Tramèr et al., 2020), highlighting the need for provable robustness guarantees.

A promising method providing such guarantees for large networks is Randomized Smoothing (RS) (Cohen et al., 2019). The core idea is to provide probabilistic robustness guarantees with arbitrarily high confidence by adding noise to the input of a base classifier and computing the expected classification over the perturbed inputs using Monte Carlo sampling. The key to obtaining high robust accuracies is a base classifier that remains consistently accurate even under high levels of noise, i.e., has low variance with respect to these perturbations. Existing works use different regularization and loss terms to encourage such behavior (Salman et al., 2019; Zhai et al., 2020; Jeong & Shin, 2020) but are ultimately all limited by the bias-variance trade-off of individual models. We show first theoretically and then empirically how ensembles can be constructed to significantly reduce this variance component and thereby increase certified radii for individual samples, and consequently, certified accuracy across the whole dataset. We illustrate this in Fig. 1.

Ensembles are a well-known tool for reducing classifier variance at the cost of increased computational cost (Hansen & Salamon, 1990). However, in the face of modern architectures (He et al., 2016; Huang et al., 2017) allowing the stable training of large models, they have been considered computationally inefficient. Yet, recent work (Wasay & Idreos, 2021) shows ensembles to be more efficient than single monolithic networks in many regimes. In light of this, we develop a theoretical framework analyzing this variance reducing property of ensembles under the perturbations introduced by RS. Further, we show how this reduced variance can significantly increase the majority class's prediction probability, leading to much larger certified radii than evaluating more perturbations with an individual model.

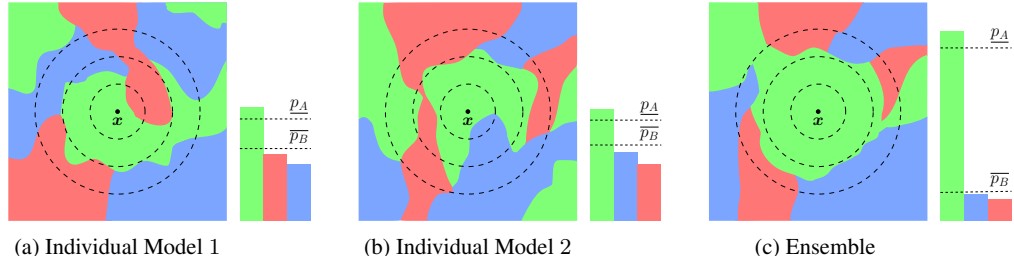

(a) Individual Model 1       (b) Individual Model 2       (c) Ensemble

Figure 1: Illustration of the prediction landscape of base models $F$ where colors represent classes. The bars show the class probabilities of the corresponding smoothed classifiers $\mathbb{E}_{\epsilon \sim \mathcal{N}(0, \sigma_\epsilon \boldsymbol{I})} F(\boldsymbol{x} + \epsilon)$, i.e., the Gaussian weighted average around $\boldsymbol{x}$ (dashed circles show the level sets of $\mathcal{N}(\boldsymbol{x}, \sigma_\epsilon \boldsymbol{I})$). The individual models in (a) and (b) predict the same class for $\boldsymbol{x}$ as their ensemble in (c). However, the ensemble's lower bound on the majority class' probability $\underline{p_A}$ is increased while its upper bound on the runner-up class' probability $\overline{p_B}$ is decreased, leading to improved robustness through RS.

Certification with RS is computationally costly as the base classifier has to be evaluated many thousands of times. To avoid exacerbating these costs by using ensembles as base models, we develop two techniques: (i) an adaptive sampling scheme for RS, which certifies samples for predetermined certification radii in stages, reducing the mean certification time up to 55-fold, and (ii) a special aggregation mechanism for ensembles which only evaluates the full ensemble on challenging samples, for which there is no consensus between a predefined subset of the constituting models.

**Main Contributions** Our key contributions are:

- A novel, theoretically motivated, and statistically sound soft-ensemble scheme for Randomized Smoothing, reducing perturbation variance and increasing certified radii (§4 and §5).

- A data-dependent adaptive sampling scheme for RS that reduces the sample complexity for predetermined certification radii in a statistically sound manner (§6).

- An extensive evaluation, examining the effects and interactions of ensemble size, training method, and perturbation size. We obtain state-of-the-art results on ImageNet and CIFAR10 for a wide range of settings, including denoised smoothing (§7).

## 2 RELATED WORK

**Adversarial Robustness** Following the discovery of adversarial examples (Biggio et al., 2013; Szegedy et al., 2014), adversarial defenses aiming to robustify networks were proposed (Madry et al., 2018). Particularly relevant to this work are approaches that certify or enforce robustness properties. We distinguish deterministic and probabilistic methods. Deterministic certification methods compute the reachable set for given input specifications using convex relaxations (Singh et al., 2019; Xu et al., 2020), mixed-integer linear programming (Tjeng et al., 2019), semidefinite programming (Dathathri et al., 2020), or SMT (Ehlers, 2017), to reason about properties of the output. To obtain networks amenable to such approaches, specialized training methods have been proposed (Mirman et al., 2018; Balunovic & Vechev, 2020; Xu et al., 2020). Probabilistic certification (Li et al., 2019; Lécuyer et al., 2019) introduces noise to the classification process to obtain probabilistic robustness guarantees, allowing the certification of larger models than deterministic methods. We review Randomized Smoothing (RS) (Cohen et al., 2019) in §3 and associated training methods (Jeong & Shin, 2020; Zhai et al., 2020; Salman et al., 2019) in App. G.3. Orthogonally to training, RS has been extended in numerous ways (Lee et al., 2019; Dvijotham et al., 2020), which we review in App. B.

**Ensembles** Ensembles have been extensively analyzed with respect to different aggregation methods (Kittler et al., 1998; Inoue, 2019), diversification (Dietterich, 2000), and the reduction of generalization errors (Tumer & Ghosh, 1996a;b). Randomized Smoothing and ensembles were first combined in Liu et al. (2020) as ensembles of smoothed classifiers. However, the method does not retain strong certificates for individual inputs; thus, we consider the work to be in a different setting from ours. We discuss this in App. A. While similar at first glance, Qin et al. (2021) randomly sample models to an ensemble, evaluating them under noise to obtain an empirical defense against adversarial attacks. They, however, do not provide robustness guarantees.

## 3 RANDOMIZED SMOOTHING

Here, we review the relevant background on Randomized Smoothing (RS) as introduced in Cohen et al. (2019). We let $f\colon \mathbb{R}^d \mapsto \mathbb{R}^m$ denote a base classifier that takes a $d$-dimensional input and produces $m$ numerical scores (pre-softmax logits), one for each class. Further, we let $F(\boldsymbol{x}) := \arg\max_q f_q(\boldsymbol{x})$ denote a function $\mathbb{R}^d \mapsto [1, \ldots, m]$ that directly outputs the class with the highest score.

For a random variable $\epsilon \sim \mathcal{N}(0, \sigma_\epsilon^2 \boldsymbol{I})$, we define a smoothed classifier $G\colon \mathbb{R}^d \mapsto [1, \ldots, m]$ as

$$G(\boldsymbol{x}) := \arg\max_c \mathcal{P}_{\epsilon \sim \mathcal{N}(0, \sigma_\epsilon^2 \boldsymbol{I})}(F(\boldsymbol{x} + \epsilon) = c). \tag{1}$$

This classifier $G$ is then robust to adversarial perturbations as follows:

**Theorem 3.1** (From Cohen et al. (2019)). *Let $c_A \in [1, \ldots, m]$, $\underline{p_A}, \overline{p_B} \in [0, 1]$. If*

$$\mathcal{P}_\epsilon(F(\boldsymbol{x} + \epsilon) = c_A) \geq \underline{p_A} \geq \overline{p_B} \geq \max_{c \neq c_A} \mathcal{P}_\epsilon(F(\boldsymbol{x} + \epsilon) = c),$$

*then $G(\boldsymbol{x} + \delta) = c_A$ for all $\delta$ satisfying $\|\delta\|_2 < R$ with $R := \frac{\sigma_\epsilon}{2}(\Phi^{-1}(\underline{p_A}) - \Phi^{-1}(\overline{p_B}))$.*

Here, $\Phi^{-1}$ denotes the inverse Gaussian CDF. Computing the exact probabilities $\mathcal{P}_\epsilon(F(\boldsymbol{x} + \epsilon) = c)$ is generally intractable. Thus, to allow practical application, CERTIFY (Cohen et al., 2019) (see Algorithm 1) utilizes sampling: First, $n_0$ samples to determine the majority class, then $n$ samples to compute a lower bound $\underline{p_A}$ to the success probability with confidence $1 - \alpha$ via the Clopper-Pearson lemma (Clopper & Pearson, 1934). If $\underline{p_A} > 0.5$, we set $\overline{p_B} = 1 - \underline{p_A}$ and

---

**Algorithm 1** Certify from (Cohen et al., 2019)

1: **function** CERTIFY($F, \sigma_\epsilon, x, n_0, n, \alpha$)
2:  cnts0 ← SAMPLEUNDERNOISE($F, x, n_0, \sigma_\epsilon$)
3:  $\hat{c}_A$ ← top index in cnts0
4:  cnts ← SAMPLEUNDERNOISE($F, x, n, \sigma_\epsilon$)
5:  $\underline{p_A}$ ← LOWERCONFBND(cnts[$\hat{c}_A$], $n, 1 - \alpha$)
6:  **if** $\underline{p_A} > \frac{1}{2}$ **then**
7:    **return** prediction $\hat{c}_A$ and radius $\sigma_\epsilon \Phi^{-1}(\underline{p_A})$
8:  **return** ⊘

---

obtain radius $R = \sigma_\epsilon \Phi^{-1}(\underline{p_A})$ via Theorem 3.1, else we abstain (return ⊘). See App. F for exact definitions of the sampling and lower bounding procedures.

To obtain high certified radii, the base model $F$ has to be trained to cope with the added Gaussian noise $\epsilon$. To achieve this, several training methods, discussed in App. G.3, have been introduced.

We also see this in Fig. 1, where various models obtain different $\underline{p_A}$, and thereby different radii $R$.

## 4 RANDOMIZED SMOOTHING FOR ENSEMBLE CLASSIFIERS

In this section, we extend the methods discussed in §3 from single models to ensembles.

For a set of $k$ classifiers $\{f^l\colon \mathbb{R}^d \mapsto \mathbb{R}^m\}_{l=1}^k$, we construct an ensemble $\bar{f}$ via weighted aggregation, $\bar{f}(\boldsymbol{x}) = \sum_{l=1}^k w^l \gamma(f^l(\boldsymbol{x}))$, where $w^l$ are the weights, $\gamma\colon \mathbb{R}^m \mapsto \mathbb{R}^m$ is a post-processing function, and $f^l(\boldsymbol{x})$ are the pre-softmax outputs of an individual model. Soft-voting (where $\gamma$ denotes identity) and equal weights $w^l = \frac{1}{k}$ perform experimentally well (see App. H.3.1) while being mathematically simple. Thus, we consider soft-ensembling via the averaging of the logits:

$$\bar{f}(\boldsymbol{x}) = \frac{1}{k} \sum_{l=1}^k f^l(\boldsymbol{x}) \tag{2}$$

The ensemble $\bar{f}$ and its corresponding hard-classifier $\bar{F}(\boldsymbol{x}) := \arg\max_q \bar{f}_q(\boldsymbol{x})$, can be used without further modification as base classifiers for RS. We find that classifiers of identical architecture and trained with the same method but different random seeds are sufficiently diverse to, when ensembled with $k \in [3, 50]$, exhibit a notably reduced variance with respect to the perturbations $\epsilon$. As we will show in the next section, this increases both the true majority class probability $p_A$ and its lower confidence bound $\underline{p_A}$, raising the certified radius as per Theorem 3.1.

## 5 VARIANCE REDUCTION VIA ENSEMBLES FOR RANDOMIZED SMOOTHING

We now show how ensembling even similar classifiers $f^l$ (cf. Eq. (2)) reduces the variance over the perturbations introduced in RS significantly, thereby increasing the majority class probability $p_A$ of

resulting ensemble $\bar{f}$ and making it particularly well-suited as a base classifier for RS. To this end, we first model network outputs with *general distributions* before investigating our theory empirically for Gaussian distributions. We defer algebra and empirical justifications to App. C and D, respectively.

**Individual Classifier**  We consider individual classifiers $f^l \colon \mathbb{R}^d \mapsto \mathbb{R}^m$ and perturbed inputs $\boldsymbol{x} + \epsilon$ for a *single* arbitrary but fixed $\boldsymbol{x}$ and Gaussian perturbations $\epsilon \sim \mathcal{N}(0, \sigma_\epsilon^2 \boldsymbol{I})$. We model the pre-softmax logits $f^l(\boldsymbol{x}) =: \boldsymbol{y}^l \in \mathbb{R}^m$ as the sum of two random variables $\boldsymbol{y}^l = \boldsymbol{y}_p^l + \boldsymbol{y}_c^l$. Here, $\boldsymbol{y}_c^l$ corresponds to the classifier's behavior on the unperturbed sample and models the stochasticity in weight initialization and training with random noise augmentation. $\boldsymbol{y}_p^l$ describes the effect of the random perturbations $\epsilon$ applied during RS. Note that this split will become essential when analyzing the ensembles. We drop the superscript $l$ when discussing an individual classifier to avoid clutter.

We model the distribution of the clean component $\boldsymbol{y}_c$ over classifiers with mean $\boldsymbol{c} = \mathbb{E}_l[\boldsymbol{f}^l(\boldsymbol{x})]$, the expectation for a fixed sample $\boldsymbol{x}$ over the randomness in the training process, and covariance $\Sigma_c \in \mathbb{R}^{m \times m}$, characterizing this randomness. We assume the distribution of the perturbation effect $\boldsymbol{y}_p$ to be zero-mean (following from local linearization and zero mean perturbations) and to have covariance $\Sigma_p \in \mathbb{R}^{m \times m}$. While $\Sigma_p$ might depend on the noise level $\sigma_\epsilon$, it is distinct from it. We do not restrict the structure of either covariance matrix and denote $\Sigma_{ii} = \sigma_i^2$ and $\Sigma_{ij} = \sigma_i \sigma_j \rho_{ij}$, for standard deviations $\sigma_i$ and correlations $\rho_{ij}$. As $\boldsymbol{y}_c$ models the global training effects and $\boldsymbol{y}_p$ models the local behavior under small perturbations, we assume them to be independent. We thus obtain logits $\boldsymbol{y}$ with mean $\boldsymbol{c}$, and covariance matrix $\Sigma = \Sigma_c + \Sigma_p$.

A classifier's prediction $F^l(\boldsymbol{x}) = \arg\max_q y_q$ is not determined by the absolute values of its logits but rather by the differences between them. Thus, to analyze the classifier's behavior, we consider the differences between the majority class logit and others, referring to them as classification margins. During certification with RS, the first step is to determine the majority class. Without loss of generality, let the class with index 1 be the majority class, i.e., $A = 1$ in Theorem 3.1, leading to the classification margin $z_i = y_1 - y_i$. Note that if $z_i > 0$ for all $i \neq 1$, then the majority class logit $y_1$ is larger than the logits of all other classes $y_i$. Under the above assumptions, the classification margin's statistics for an individual classifier are:

$$\mathbb{E}[z_i] = c_1 - c_i$$
$$\mathrm{Var}[z_i] = \sigma_{p,1}^2 + \sigma_{p,i}^2 + \sigma_{c,1}^2 + \sigma_{c,i}^2 - 2\rho_{p,1i}\sigma_{p,1}\sigma_{p,i} - 2\rho_{c,1i}\sigma_{c,1}\sigma_{c,i}.$$

**Ensemble**  Now, we construct an ensemble of $k$ such classifiers. We use soft-voting (cf. Eq. (2)) to compute the ensemble output $\bar{\boldsymbol{y}} = \frac{1}{k}\sum_{l=1}^k \boldsymbol{y}^l$ and then the corresponding classification margins $\bar{z}_i = \bar{y}_1 - \bar{y}_i$. We consider similar classifiers, differing only in the random seed used for training. Hence, we assume that the correlation between the logits of different classifiers has a similar structure but smaller magnitude than the correlation between logits of one classifier. Correspondingly, we parametrize the covariance between $\boldsymbol{y}_c^i$ and $\boldsymbol{y}_c^j$ for classifiers $i \neq j$ with $\zeta_c \Sigma_c$ and similarly between $\boldsymbol{y}_p^i$ and $\boldsymbol{y}_p^j$ with $\zeta_p \Sigma_p$ for $\zeta_c, \zeta_p \in [0, 1]$. Note that, as we capture the randomness introduced in the training process with $\Sigma_c$, we use the same $\Sigma_c, \Sigma_p$ for each individual model. With these correlation coefficients $\zeta_c$ and $\zeta_p$, this construction captures the range from no correlation ($\zeta = 0$) to perfect correlation ($\zeta = 1$). By the linearity of expectation and this construction, respectively, we obtain:

$$\mathbb{E}[\bar{z}_i] = \mathbb{E}[z_i] = c_1 - c_i$$

$$\mathrm{Var}[\bar{z}_i] = \underbrace{\frac{k + 2\binom{k}{2}\zeta_p}{k^2}(\sigma_{p,1}^2 + \sigma_{p,i}^2 - 2\rho_{p,1i}\sigma_{p,1}\sigma_{p,i})}_{\sigma_p^2(k)} + \underbrace{\frac{k + 2\binom{k}{2}\zeta_c}{k^2}(\sigma_{c,1}^2 + \sigma_{c,i}^2 - 2\rho_{c,1i}\sigma_{c,1}\sigma_{c,i})}_{\sigma_c^2(k)}.$$

**Variance Reduction**  We can split $\mathrm{Var}[\bar{z}_i]$ into the components associated with the clean prediction $\sigma_c^2(k)$ and the perturbation effect $\sigma_p^2(k)$, both as functions of the ensemble size $k$. We now compare these variance terms independently to the corresponding terms of an individual classifier dropping the subscripts $p$ and $c$ from $\sigma_c^2(k)/\sigma_c^2(1)$ and $\sigma_p^2(k)/\sigma_p^2(1)$ as they follow the same structure:

$$\frac{\sigma^2(k)}{\sigma^2(1)} = \frac{(1 + \zeta(k-1))(\sigma_1^2 + \sigma_i^2 - 2\rho_{1i}\sigma_1\sigma_i)}{k(\sigma_1^2 + \sigma_i^2 - 2\rho_{1i}\sigma_1\sigma_i)} = \frac{1 + \zeta(k-1)}{k} \xrightarrow{k \to \infty} \zeta \qquad (3)$$

We observe that both variance component ratios go towards their corresponding correlation coefficients $\zeta_c$ and $\zeta_p$ as ensemble size grows. As we now proceed to show (see Fig. 2a, explained later), this corresponds to an increase in success probability $p_1$ and thereby certified radius.

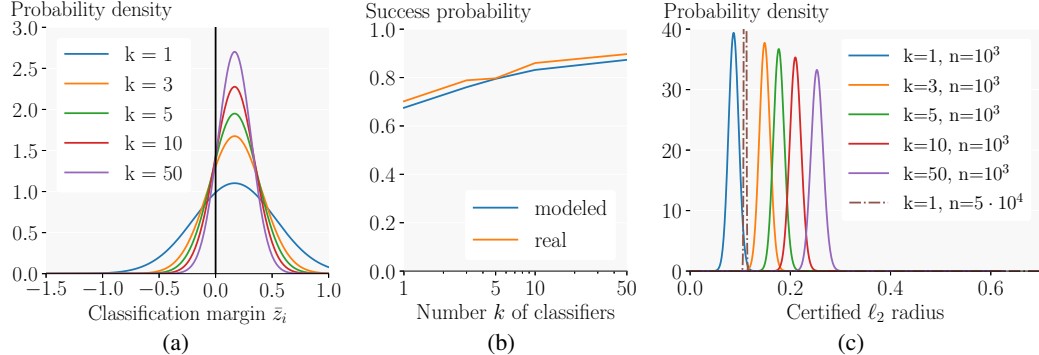

Figure 2: Comparison of the (a) modeled distribution over the classification margin $z_i$ to the runner-up class as a Gaussian, (b) the resulting success probability $p_1$ of predicting the majority class 1 for a given perturbation $\epsilon$ modeled via Gaussian and observed via sampling ("real"), and (c) the corresponding distribution over certified radii for ensembles of different sizes.

Our modeling approach is mathematically closely related to Tumer & Ghosh (1996a), which focuses on analyzing ensemble over the whole dataset. However, we condition on a single input to study the interplay between the stochasticity in training and random perturbations encountered in RS.

**Effect on Success Probability**    We can compute the probability of an ensemble of $k$ classifiers predicting the majority class by integrating the probability distribution of the classification margin over the orthant[1] $\bar{z}_i > 0 \; \forall i \geq 2$ where majority class 1 is predicted:

$$p_1 := \mathcal{P}(\bar{F}(\boldsymbol{x} + \epsilon) = 1) = \mathcal{P}\left(\bar{z}_i > 0 : \forall \, 2 \leq i \leq m\right) = \int_{\substack{\bar{z} \, s.t. \, \bar{z}_i > 0, \\ \forall \, 2 \, \leq i \leq m}} \mathcal{P}(\bar{z}) \, d\bar{z}. \tag{4}$$

Assuming Gaussian distributions for $\boldsymbol{z}$, we observe the increase in success probability shown in Fig. 2b. Without assuming a specific distribution of $\boldsymbol{z}$, we can still lower-bound the success probability using Chebyshev's inequality and the union bound. Given that a mean classification yields the majority class and hence $c_1 - c_i > 0$, we let $t_i = \frac{c_1 - c_i}{\sigma_i(k)}$ and have:

$$p_1 \geq 1 - \sum_{i=2}^{m} \mathcal{P}\left(|\bar{z}_i - c_1 + c_i| \geq t_i \, \sigma_i(k)\right) \geq 1 - \sum_{i=2}^{m} \frac{\sigma_i(k)^2}{(c_1 - c_i)^2} \tag{5}$$

where $\sigma_i(k)^2 = \sigma_{c,i}(k)^2 + \sigma_{p,i}(k)^2$ is the variance of classification margin $\bar{z}_i$. We observe that as $\sigma_i(k)$ decreases with increasing ensemble size $k$ (see Eq. (3)), the lower bound to the success probability approaches 1 quadratically. The further we are away from the decision boundary, i.e., the larger $c_i$, the smaller the absolute increase in success probability for the same variance reduction.

Given a concrete success probability $p_1$, we compute the probability distribution over the certifiable radii reported by CERTIFY (Algorithm 1) for a given confidence $\alpha$, sample number $n$, and perturbation variance $\sigma_\epsilon^2$ (up to choosing an incorrect majority class $\hat{c}_A$) as:

$$\mathcal{P}\left(R = \sigma_\epsilon \Phi^{-1}\left(\underline{p_1}(n_1, n, \alpha)\right)\right) = \mathcal{B}(n_1, n, p_1), \quad \text{for } R > 0 \tag{6}$$

where $\mathcal{B}(s, r, p)$ is the probability of drawing $s$ successes in $r$ trials from a Binomial distribution with success probability $p$, and $\underline{p}(s, r, \alpha)$ is the lower confidence bound to the success probability of a Bernoulli experiment given $s$ successes in $r$ trials with confidence $\alpha$ according to the Clopper-Pearson interval (Clopper & Pearson, 1934). We illustrate the resulting effect assuming Gaussian distributions in Fig. 2c.

**Empirical Analysis via Gaussian Assumption**    To investigate our theory, we now assume $\boldsymbol{y}_c^l$ and $\boldsymbol{y}_p^l$ and hence also $\bar{z}_i$ to be multivariate Gaussians. This choice is empirically well-fitting (see App. D) and follows from the central limit theorem for $\boldsymbol{y}_c^l$ and from Gaussian perturbations and local

---

[1]An orthant is the n-dimensional equivalent of a quadrant.

linearization for $y_p^l$. To estimate the free parameters $c$, $\Sigma_c$, $\Sigma_p$, $\zeta_c$, and $\zeta_p$, we evaluate ensembles of up to $k = 50$ GAUSSIAN trained `ResNet20` (for details, see §7) at $\sigma_\epsilon = 0.25$. We obtain $c$ and $\Sigma_c$ as the mean and covariance of the output on a randomly chosen sample $x$. Subtracting the clean outputs from those for the perturbed samples, we estimate the covariance matrix $\Sigma_p$. We determine $\zeta_c \approx 0$ and $\zeta_p \approx 0.82$ as the median ratio of the inter- and intra-classifier covariance. $\zeta_c \approx 0$ implies that our models can be treated as independent conditioned on a single fixed and unperturbed input.

Plugging these estimates into our construction under the Gaussian assumption, we observe a significant decrease in the variance of the classification margin to the runner-up class as ensemble size $k$ increases (see Fig. 2a). This generally leads to more polarized success probabilities: the majority class' probability is increased, while the other classes' probabilities are decreased because the probability mass concentrates around the mean, and consequently on one side of the decision threshold $\bar{z} = 0$, which determines the success probability via Eq. (4). An increase, as in our case for a correct mean prediction (see Fig. 2b), leads to much larger expected certified radii (see Fig. 2c) for a given number of sampled perturbations (here $n = 10^3$) via Eq. (6). In contrast, sampling more perturbations will, in the limit, only recover the true success probability. In our example, going from one classifier to an ensemble of 50 increases the expected certified radius by $191\%$, while drawing 50 times more

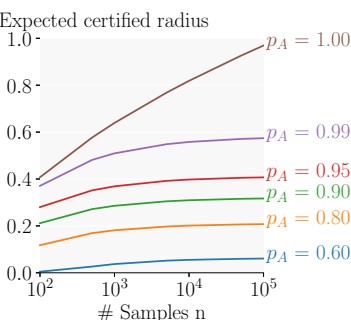

Figure 3: Expected certified radius over the number $n$ of samples for the true success probabilities $p_A$.

perturbations for a single model only yields a $28\%$ increase (see Fig. 2c). As illustrated in Fig. 3, these effects are strongest close to decision boundaries ($p_A \ll 1$), where small increases in $p_A$ impact the certified radius much more than the number of samples $n$, which dominated at $p_A \approx 1$.

## 6 COMPUTATIONAL OVERHEAD REDUCTION

To avoid exacerbating the high computational cost of certification via RS with the need to evaluate many models constituting an ensemble, we propose two orthogonal and synergizing approaches to reduce sample complexity significantly. First, an adaptive sampling scheme reducing the average number of samples required for certification. Second, a more efficient aggregation mechanism for ensembles, reducing the number of models evaluated per sample, also applicable to inference.

**Adaptive Sampling** We assume a setting where a target certification radius $r$ is known a priori, as is the standard in deterministic certification, and aim to show $R \geq r$ via Theorem 3.1. After choosing the majority class $\hat{c}_A$ as in CERTIFY, CERTIFYADP (see Algorithm 2) leverages the insight that relatively few samples are often sufficient to certify robustness at radius $r$. We perform an $s \geq 1$ step procedure: At step $i$, we evaluate $n_i$ fresh samples and, depending on these, either certify radius $r$ as in CERTIFY, abort certification if an upper confidence bound suggests that certification will not be fruitful, or continue with $n_{i+1} \gg n_i$. After $s$ steps, we abstain. To obtain the same strong statistical guarantees as with CERTIFY, it is essential to correct for multiple testing by applying Bonferroni correction (Bonferroni, 1936), i.e., use confidence $1 - \frac{\alpha}{s}$ rather than $1 - \alpha$. The following theorem summarizes the key properties of the algorithm:

**Theorem 6.1.** *For $\alpha, \beta \in [0, 1], s \in \mathbb{N}^+, n_1 < \cdots < n_s$, CERTIFYADP:*

1. *returns $\hat{c}_A$ if at least $1 - \alpha$ confident that $G$ is robust with a radius of at least $r$.*

2. *returns $\oslash$ before stage $s$ only if at least $1 - \beta$ confident that robustness of $G$ at radius $r$ can not be shown.*

3. *for $n_s \geq \lceil n(1 - \log_\alpha(s)) \rceil$ has maximum certifiable radii at least as large as CERTIFY for $n$.*

We provide a proof in App. E. For well-chosen parameters, this allows certifying radius $r$ with, in expectation, significantly fewer evaluations of $F$ than CERTIFY as often $n_i \ll n$ samples suffice to certify or to show that certification will not succeed. At the same time, we choose $n_s$ in accordance with (3) in Theorem 6.1 to retain the ability to certify large radii (compared to CERTIFY) despite the increased confidence level required due to Bonferroni correction. For an example, see Fig. 4.

**Algorithm 2** Adaptive Sampling

1: **function** CERTIFYADP($F, \sigma_\epsilon, \boldsymbol{x}, n_0, \{n_j\}_{j=1}^s, s, \alpha, \beta, r$)
2: $\quad$ cnts0 $\leftarrow$ SAMPLEUNDERNOISE($F, \boldsymbol{x}, n_0, \sigma_\epsilon$)
3: $\quad \hat{c}_A \leftarrow$ top index in cnts0
4: $\quad$ **for** $i \leftarrow 1$ to $s$ **do**
5: $\quad\quad$ cnts $\leftarrow$ SAMPLEUNDERNOISE($F, \boldsymbol{x}, n_i, \sigma_\epsilon$)
6: $\quad\quad \underline{p_A} \leftarrow$ LOWERCONFBND(cnts[$\hat{c}_A$], $n_i, 1 - \frac{\alpha}{s}$)
7: $\quad\quad$ **If** $\sigma_\epsilon \, \Phi^{-1}(\underline{p_A}) \geq r$ **then return** $\hat{c}_A$
8: $\quad\quad \overline{p_A} \leftarrow$ UPPERCONFBND(cnts[$\hat{c}_A$], $n_i, 1 - \frac{\beta}{s-1}$)
9: $\quad\quad$ **If** $\sigma_\epsilon \, \Phi^{-1}(\overline{p_A}) < r$ **then return** $\oslash$
10: $\quad$ **return** $\oslash$

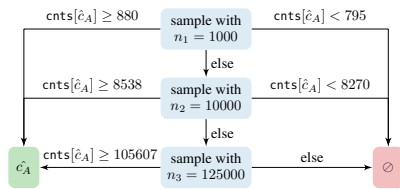

Figure 4: CERTIFYADP with $\beta = 0.0001$ obtains similar guarantee as CERTIFY at $n = 100'000$ while being more sample efficient for most inputs. Both use $\alpha = 0.001$, $\sigma_\epsilon = r = 0.25$.

**$K$-Consensus Aggregation** To reduce both inference and certification times, we can adapt the ensemble aggregation process to return an output early when there is consensus. Concretely, first, we order the classifiers in an ensemble by their accuracy (under noisy inputs) on a holdout dataset. Then, when evaluating $\bar{f}$, we query classifiers in this order. If the first $K$ individual classifiers agree on the predicted class, we perform soft-voting on these and return the result without evaluating the remaining classifiers. Especially for large ensembles, this approach can significantly reduce inference time without hampering performance, as the ratio $\frac{k}{K}$ can be large.

We note that similar approaches for partial ensemble evaluation have been proposed (Inoue, 2019; Wang et al., 2018; Soto et al., 2016), and that this does not affect the mathematical guarantees of RS.

## 7 EXPERIMENTAL EVALUATION

In line with prior work, we evaluate the proposed methods on the CIFAR10 (Krizhevsky et al., 2009) and ImageNet (Russakovsky et al., 2015) datasets with respect to two key metrics: (i) the certified accuracy at predetermined radii $r$ and (ii) the average certified radius (ACR). We show that on both datasets, all ensembles outperform their strongest constituting model and obtain a new state-of-the-art. On CIFAR10, we demonstrate that ensembles outperform individual models even when correcting for computational cost via model size or the number of used perturbations. Further, we find that using adaptive sampling and K-Consensus aggregation speeds up certification up to 55-fold for ensembles and up to 33-fold for individual models.

**Experimental Setup** We implement our approach in PyTorch (Paszke et al., 2019) and evaluate on CIFAR10 with ensembles of `ResNet20` and `ResNet110` and on ImageNet with ensembles of `ResNet50` (He et al., 2016), using 1 and 2 GeForce RTX 2080 Ti, respectively. Due to the high computational cost of CERTIFY, we follow previous work (Cohen et al., 2019) and evaluate every $20^{th}$ image of the CIFAR10 test set and every $100^{th}$ of the ImageNet test set (500 samples total).

**Training and Certification** We train models with GAUSSIAN (Cohen et al., 2019), CONSISTENCY (Jeong & Shin, 2020), and SMOOTHADV (Salman et al., 2019) training and utilize pre-trained SMOOTHADV and MACER (Zhai et al., 2020) models. More details are provided in App. G. If not declared differently, we use $n_0 = 100$, $n = 100'000$, $\alpha = 0.001$, no CERTIFYADP or $K$-Consensus and evaluate and train with the same $\sigma_\epsilon$.

### 7.1 MAIN RESULTS

**Results on CIFAR10** In Table 1, we compare ensembles of 10 `ResNet110` against individual networks at $\sigma_\epsilon = 0.50$ w.r.t. average certified radius (ACR) and certified accuracy at various radii. We consistently observe that ensembles outperform their constituting models by up to 21% in ACR and 45% in certified accuracy, yielding a new state-of-the-art at every radius. Improvements are more pronounced for larger ensembles (see Fig. 5) and at larger radii, where larger $\underline{p_A}$ are required, which agrees well with our theoretical observations in §5. We present more extensive results in App. H.1, including experiments for $\sigma_\epsilon = 0.25$ and $\sigma_\epsilon = 1.0$ yielding new state-of-the-art ACR in all settings.

Table 1: CIFAR10 average certified radius (ACR) and certified accuracy at different radii for ensembles of $k$ `ResNet110` ($k = 1$ are individual models) at $\sigma_\epsilon = 0.5$. Larger is better.

| Training | $k$ | ACR | Radius r | | | | | | | |
|---|---|---|---|---|---|---|---|---|---|---|
| | | | 0.0 | 0.25 | 0.50 | 0.75 | 1.00 | 1.25 | 1.50 | 1.75 |
| GAUSSIAN | 1 | 0.535 | 65.8 | 54.2 | 42.2 | 32.4 | 22.0 | 14.8 | 10.8 | 6.6 |
| | 10 | 0.648 | **69.0** | **60.4** | **49.8** | 40.0 | 29.8 | 19.8 | 15.0 | 9.6 |
| CONSISTENCY | 1 | 0.708 | 63.2 | 54.8 | 48.8 | 42.0 | 36.0 | 29.8 | 22.4 | 16.4 |
| | 10 | **0.756** | 65.0 | 59.0 | 49.4 | **44.8** | 38.6 | 32.0 | 26.2 | 19.8 |
| SMOOTHADV | 1 | 0.707 | 52.6 | 47.6 | 46.0 | 41.2 | 37.2 | 31.8 | 28.0 | 23.4 |
| | 10 | 0.730 | 52.4 | 48.6 | 45.8 | 42.6 | **38.8** | **34.4** | **30.4** | **25.0** |
| MACER | 1 | 0.668 | 62.4 | 54.4 | 48.2 | 40.2 | 33.2 | 26.8 | 19.8 | 13.0 |

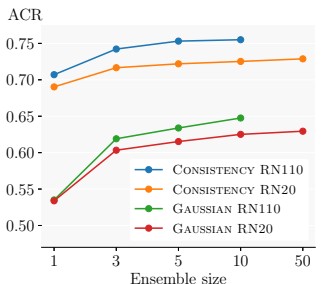

ACR

Figure 5: ACR over ensemble size $k$ for $\sigma_\epsilon = 0.5$ on CIFAR10.

Table 2: ImageNet average certified radius (ACR) and certified accuracy at different radii for ensembles of $k$ `ResNet50` ($k = 1$ are individual models) at $\sigma_\epsilon = 1.0$. Larger is better.

| Training | $k$ | ACR | Radius r | | | | | | | |
|---|---|---|---|---|---|---|---|---|---|---|
| | | | 0.0 | 0.50 | 1.00 | 1.50 | 2.00 | 2.50 | 3.00 | 3.50 |
| GAUSSIAN | 1 | 0.875 | 43.6 | 37.8 | 32.6 | 26.0 | 19.4 | 14.8 | 12.2 | 9.0 |
| | 3 | 0.968 | 43.8 | 38.4 | 34.4 | 29.8 | 23.2 | 18.2 | 15.4 | 11.4 |
| CONSISTENCY | 1 | 1.022 | 43.2 | 39.8 | 35.0 | 29.4 | 24.4 | 22.2 | 16.6 | 13.4 |
| | 3 | **1.108** | 44.6 | 40.2 | **37.2** | **34.0** | **28.6** | 23.2 | 20.2 | 16.4 |
| SMOOTHADV | 1 | 1.011 | 40.6 | 38.6 | 33.8 | 29.8 | 25.6 | 20.6 | 18.0 | 14.4 |
| | 3 | 1.065 | 38.6 | 36.0 | 34.0 | 30.0 | 27.6 | **24.6** | **21.2** | **18.8** |
| MACER † | 1 | 1.008 | **48** | **43** | 36 | 30 | 25 | 18 | 14 | - |

† As reported by Zhai et al. (2020).

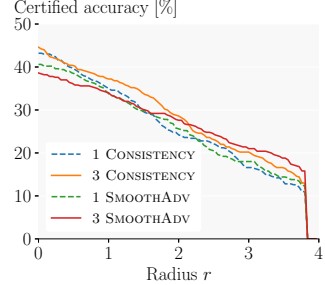

Certified accuracy [%]

Figure 6: Certified acc. over radius $r$ for $\sigma_\epsilon = 1.0$ on ImageNet.

**Results on ImageNet** In Table 2, we compare ensembles of 3 `ResNet50` with individual models at $\sigma_\epsilon = 1.0$, w.r.t. ACR and certified accuracy. We observe similar trends to CIFAR10, with ensembles outperforming their constituting models and the ensemble of 3 CONSISTENCY trained `ResNet50` obtaining a new state-of-the-art ACR of $1.11$, implying that ensembles are effective on a wide range of datasets. We visualize this in Fig. 6 and present more detailed results in App. H.2, where we also provide experiments for $\sigma_\epsilon = 0.25$ and $\sigma_\epsilon = 0.5$, yielding new state-of-the-art ACR in both cases.

**Computational Overhead Reduction** We evaluate CERTIFYADP in conjunction with $K$-Consensus aggregation using $\beta = 0.001$, $\{n_j\} = \{100, 1'000, 10'000, 120'000\}$, and $K = 5$ in Table 3 and report KCR, the percentage of inputs for which only $K$ classifiers were evaluated and SampleRF and TimeRF, the factors by which sample complexity and certification time are reduced, respectively. We observe up to 67-fold certification speed-ups and up to 55-fold sample complexity

Table 3: Adaptive sampling and $K$-Consensus aggregation on CIFAR10 for 10 CONSISTENCY trained `ResNet110`.

| Radius | $\sigma_\epsilon$ | $\mathrm{acc_{cert}}$ [%] | SampleRF | KCR [%] | TimeRF |
|---|---|---|---|---|---|
| 0.25 | 0.25 | 70.4 | **55.24** | 39.3 | **67.16** |
| 0.50 | 0.25 | 60.6 | 18.09 | 57.3 | 25.07 |
| 0.75 | 0.25 | 52.0 | 5.68 | 87.8 | 10.06 |
| 1.00 | 0.50 | 38.6 | 14.79 | 78.3 | 24.01 |
| 1.25 | 0.50 | 32.2 | 8.14 | 92.7 | 15.06 |
| 1.50 | 0.50 | 26.2 | 6.41 | **97.5** | 12.31 |

reductions for individual predetermined radii, without incurring any accuracy penalties. This way, certifying an ensemble of $k = 10$ networks with CERTIFYADP can be significantly faster than certifying an individual model with CERTIFY while yielding notably higher certified accuracies. We observe that adaptive sampling and $K$-Consensus aggregation complement each other well: At larger radii $r$, more samples pass on to the later stages of CERTIFYADP, requiring more model evaluations. However, only samples with high success probabilities reach these later stages, making it more likely for $K$-Consensus aggregation to terminate an evaluation early. We provide extensive additional experiments in App. H.4.

## 7.2 ABLATION STUDY

**Ensemble Size and Model Size Ablation** In Fig. 5, we illustrate the effect of ensemble size on ACR for various training methods and model architectures, providing more extensive results in App. H.3.2. We observe that in all considered settings, even relatively small ensembles of just 3-5 models realize most of the improvements to be made. In fact, as few as 3 and 5 `ResNet20` are enough to outperform a single `ResNet110` under GAUSSIAN and CONSISTENCY training, respectively, while only having $47\%$ and $79\%$ of the parameters.

Table 4: Adaptive sampling for different $\{n_j\}$ on CIFAR10 at $\sigma_\epsilon = 0.25$. SampleRF and TimeRF are the reduction factors compared to standard sampling with $n = 100'000$ (larger is better). $\text{ASR}_j$ is the portion of certification attempts returned in phase $j$.

| $r$ | Training | $\{n_j\}$ | $\text{acc}_{\text{cert}}$ [%] | $\text{ASR}_1$ [%] | $\text{ASR}_2$ [%] | $\text{ASR}_3$ [%] | $\text{ASR}_4$ [%] | SampleRF | TimeRF |
|---|---|---|---|---|---|---|---|---|---|
| 0.25 | GAUSSIAN | $1'000, 110'000$ | 60.0 | 92.2 | 7.8 | - | - | 10.34 | 10.45 |
| | | $1'000, 10'000, 116'000$ | 60.0 | 91.6 | 5.0 | 3.4 | - | 17.01 | 17.21 |
| | | $100, 1'000, 10'000, 120'000$ | 60.0 | 75.4 | 16.0 | 5.6 | 3.0 | 20.40 | 21.22 |
| | CONSISTENCY | $100, 1'000, 10'000, 120'000$ | 66.0 | 84.8 | 10.8 | 2.6 | 1.8 | **33.91** | **34.67** |
| 0.75 | GAUSSIAN | $100, 1'000, 10'000, 120'000$ | 30.2 | 48.8 | 11.6 | 29.4 | 10.2 | **5.92** | **6.10** |
| | CONSISTENCY | $100, 1'000, 10'000, 120'000$ | 46.4 | 32.4 | 8.2 | 49.4 | 10.0 | 5.32 | 5.40 |

**Equal Number of Inferences** In Fig. 7, we compare the effect of evaluating a larger number of perturbations $n$ for an individual model compared to an ensemble at the same computational cost. We observe that increasing the sample count for a single model only enables marginal improvements at most radii and only outperforms the ensemble at radii, which can mathematically not be certified using the smaller sample count. See also Fig. 3 for a theoretical explanation and App. H.3.4 for more experiments.

**Other Base Classifiers** While our approach works exceptionally well when ensembling similar classifiers, it can also be successfully extended to base models trained with different training methods (see App. H.3.5) and denoised classifiers (see App. H.3.6), yielding ACR improvements of up to $5\%$ and $17\%$ over their strongest constituting models, respectively. This way, existing models can be reused to avoid the high cost of training an ensemble.

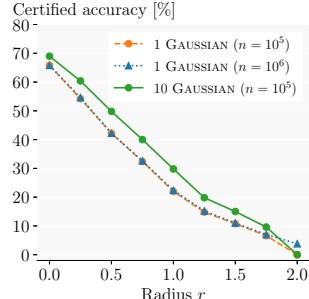

Figure 7: Comparing an ensemble and an individual model at an equal number of total inferences on CIFAR10.

**Adaptive Sampling Ablation** In Table 4, we show the effectiveness of CERTIFYADP for certifying robustness at predetermined radii using $\beta = 0.001$, different target radii, training methods, and sampling count sets $\{n_j\}$. We compare $\text{ASR}_j$, the portion of samples returned in stage $j$, SampleRF, and TimeRF. Generally, we observe larger speed-ups for smaller radii (up to a factor of 34 at $r = 0.25$). There, certification only requires small $\underline{p_A}$, which can be obtained even with few samples. For large radii (w.r.t $\sigma_\epsilon$), higher $\underline{p_A}$ must be shown, requiring more samples even at true success rates of $p_A = 1$. There, the first phase only yields early abstentions. The more phases we use, the more often we certify or abstain early, but also the bigger the costs due to Bonferroni correction. We observe that exponentially increasing phase sizes and in particular $\{n_j\} = \{100, 1'000, 10'000, 120'000\}$ perform very well across different settings. See App. H.4.1 for additional results.

**$K$-Consensus Aggregation Ablation** In Table 5, we compare ACR and certification time reduction for ensembles of 10 `ResNet110` and 50 `ResNet20` using $K$-Consensus aggregation in isolation across different $K$. Even when using only $K = 2$, we already obtain $81\%$ and $70\%$ of the ACR improvement obtainable by always evaluating the full ensembles ($k = 10$ and $k = 50$), respectively. The more conservative $K = 10$ for `ResNet20` still reduces certification times by a factor of 2 without losing accuracy. See App. H.4.2 for more detailed experiments.

Table 5: $K$-Consensus aggregation at $\sigma_\epsilon = 0.25$ on CIFAR10.

| Architecture | $K$ | ACR | Time$RF$ | KCR |
|---|---|---|---|---|
| CONSISTENCY ResNet110 | 1 | 0.546 | 10.0 | 100.0 |
| | 2 | 0.576 | 3.25 | 85.8 |
| | 5 | 0.583 | 1.59 | 74.2 |
| | 10 | 0.583 | 1.00 | 0.0 |
| CONSISTENCY ResNet20 | 1 | 0.528 | 50.0 | 100.0 |
| | 2 | 0.544 | 6.50 | 87.7 |
| | 10 | 0.551 | 2.01 | 69.8 |
| | 50 | 0.551 | 1.00 | 0.0 |

## 8 CONCLUSION

We propose a theoretically motivated and statistically sound approach to construct low variance base classifiers for Randomized Smoothing by ensembling. We show theoretically and empirically why this approach significantly increases certified accuracy yielding state-of-the-art results. To offset the computational overhead of ensembles, we develop a generally applicable adaptive sampling mechanism, reducing certification costs up to 55-fold for predetermined radii and an ensemble aggregation mechanism, complementing it and reducing evaluation costs on its own up to 6-fold.

## 9 ETHICS STATEMENT

Most machine learning techniques can be applied both in ethical and unethical ways. Techniques to increase the certified robustness of models do not change this but only make the underlying models more robust, again allowing beneficial and malicious applications, e.g., more robust medical models versus more robust weaponized AI. As our contributions improve certified accuracy, certification radii, and inference speed, both of these facets of robust AI are amplified. Furthermore, while we achieve state-of-the-art results, these do not yet extend to realistic perturbations. Malicious actors may aim to convince regulators that methods such as the proposed approach are sufficient to provide guarantees for general real-world threat scenarios, leading to insufficient safeguards.

## 10 REPRODUCIBILITY STATEMENT

We make all code and pre-trained models required to reproduce our results publicly available at `https://github.com/eth-sri/smoothing-ensembles`. There, we also provide detailed instructions and scripts facilitating the reproduction of our results. We explain the basic experimental setup in §7 and provide exact experimental details in App. G, where we extensively describe the datasets (App. G.1), model architectures (App. G.2), training methods (App. G.3), hyper-parameter choices (App. G.4), and experiment timings (App. G.5). We remark that the datasets, architectures, and training methods we use are all publicly available. Section §7 and App. H contain numerous experiments for various datasets, model architectures, training methods, noise levels, and ensemble compositions. The consistent trends we observe over this wide range of settings highlight the general applicability of our approach. In App. H.3.3, we analyze the variability of our results, reporting standard deviations for a range of metrics, find that our ensembling approach reduces variability, and note that our results are statistically significant. We include complete proofs of all theoretical contributions (§5 and §6) in App. C and E. In App. D, we theoretically motivate the modelling assumptions made in §5 and validate them empirically.

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

## A  COMPARISON TO LIU ET AL. (2020)

Liu et al. (2020) introduce Smoothed Weighted Ensembling (SWEEN), which are weighted ensembles of smoothed base models. They focus on deriving optimal ensemble weights for generalization rather than on analyzing the particularities of ensembles in the RS setting. As they do not consider multiple testing correction and the confidence of the applied Monte Carlo sampling, they obtain statements about empirical, distributional robustness of their classifiers, rather than individual certificates with high confidence. Due to this difference setting, we do not compare numerically.

## B  ADDITIONAL RELATED WORK

**Extensions**   As outlined in §3, Cohen et al. (2019) focuses on the addition of Gaussian noise to drive $\ell_2$-robustness results. However, various extensions with other types of noise and guarantees have been proposed.

By using different types of noise distributions and radius calculations, Yang et al. (2020); Zhang et al. (2020) derive recipes to determine certificates for general $\ell_p$-balls and specifically showcase results for $p = 1, 2, \infty$.

Numerous works have shown extensions to discrete perturbations such as $\ell_0$-perturbations (Lee et al., 2019; Wang et al., 2021; Schuchardt et al., 2021), graphs (Bojchevski et al., 2020; Gao et al., 2020), patches Levine & Feizi (2020) or manipulations of points in a point cloud Liu et al. (2021).

Dvijotham et al. (2020) provide theoretical derivations for the application of both continuous and discrete smoothing measures.

Mohapatra et al. (2020) improve certificates by considering gradient information.

Beyond norm-balls certificates, Fischer et al. (2020); Li et al. (2021) show how geometric operations such as rotation or translation can be certified via Randomized Smoothing.

Chiang et al. (2020); Fischer et al. (2021) show how the certificates can be extended from the setting of classification to regression (and object detection) and segmentation, respectively. In the classification setting, Jia et al. (2020) extend certificates from just the top-1 class to the top-k classes. Similarly, Kumar et al. (2020a) certify the confidence of the classifier not, just the top class prediction.

Beyond the inference-time evasion attacks, Rosenfeld et al. (2020) showcase RS as a defense against data poisoning attacks.

Many of these extensions are orthogonal to the standard Randomized Smoothing approach and thus orthogonal to our improvements. We showcase that these improvements carry over from the $\ell_2$ case to other $\ell_p$ norms in App. H.3.7.

**Limitations**   Recently, some works also have investigated the limitations of Randomized Smoothing: Mohapatra et al. (2021) point out the limitations of training for Randomized Smoothing as well as the impact on class-wise accuracy. Kumar et al. (2020b); Wu et al. (2021) show that for $p > 2$, the achievable certification radius quickly diminishes in high-dimensional spaces.

## C  MATHEMATICAL DERIVATION OF VARIANCE REDUCTION FOR ENSEMBLES

In this section, we present the algebraic derivations for §5, skipped in the main part due to space constraints.

**Individual Classifier**   In §5 define $f^l(\boldsymbol{x}) =: \boldsymbol{y}^l \in \mathbb{R}^m$ as the sum of two random variables $\boldsymbol{y}^l = \boldsymbol{y}_p^l + \boldsymbol{y}_c^l$.

The behavior on a specific clean sample $\boldsymbol{x}$ is modeled by $\boldsymbol{y}_c^l$ with mean $\boldsymbol{c} \in \mathbb{R}^m$, the expectation of the logits for this sample over the randomization in the training process, and corresponding covariance

$\Sigma_c$:

$$\mathbb{E}[\boldsymbol{y}_c^l] = \boldsymbol{c}$$

$$\mathrm{Cov}[\boldsymbol{y}_c^l] = \Sigma_c = \begin{bmatrix} \sigma_{c,1}^2 & \cdots & \rho_{c,1m}\sigma_{c,1}\sigma_{c,m} \\ & \ddots & \vdots \\ & & \sigma_{c,m}^2 \end{bmatrix}$$

We note that: (i) we omit the lower triangular part of covariance matrices due to symmetry, (ii) $\boldsymbol{c}$ and $\Sigma_c$ are constant across the different $f^l$ for a fixed $\boldsymbol{x}$ and (iii) that due to (ii) and our modelling assumptions all means, (co)variances and probabilities are conditioned on $\boldsymbol{x}$, e.g., $\mathbb{E}[\boldsymbol{y}_c^l] = \mathbb{E}[\boldsymbol{y}_c^l \mid \boldsymbol{x}] = \boldsymbol{c}$. However, as we define all random variables only for a fixed $\boldsymbol{x}$, we omit this.

Similarly, we model the impact of the perturbations $\epsilon$ introduced during RS by $\boldsymbol{y}_p^l$ with mean zero and covariance $\Sigma_p$:

$$\mathbb{E}[\boldsymbol{y}_p^l] = \boldsymbol{0}$$

$$\mathrm{Cov}[\boldsymbol{y}_p^l] = \Sigma_p = \begin{bmatrix} \sigma_{p,1}^2 & \cdots & \rho_{p,1m}\sigma_{p,1}\sigma_{p,m} \\ & \ddots & \vdots \\ & & \sigma_{p,m}^2 \end{bmatrix}$$

That is, we assume $\boldsymbol{c}^l$ to be the expected classification a model learns for the clean sample while the covariances $\Sigma_c$ and $\Sigma_p$ encode the stochasticity of the training process and perturbations, respectively. As $\boldsymbol{y}_p^l$ models the local behavior under small perturbations and $\boldsymbol{y}_c^l$ models the global training effects, we assume them to be independent:

$$\mathbb{E}[\boldsymbol{y}^l] = \boldsymbol{c}$$

$$\mathrm{Cov}[\boldsymbol{y}^l] = \Sigma = \Sigma_c + \Sigma_p = \begin{bmatrix} \sigma_{p,1}^2 + \sigma_{c,1}^2 & \cdots & \rho_{p,1m}\sigma_{p,1}\sigma_{p,m} + \rho_{c,1m}\sigma_{c,1}\sigma_{c,m} \\ & \ddots & \vdots \\ & & \sigma_{p,m}^2 + \sigma_{c,m}^2 \end{bmatrix}.$$

The classifier prediction $\arg\max_q y_q$ is determined by the differences between logits. We call the difference between the target logit and others the classification margin. During certification with RS, the first step is to determine the majority class. Without loss of generality, we assume that it has been determined to be index 1, leading to the classification margin $z_i = y_1 - y_i$. If $z_i > 0$ for all $i \neq 1$, the majority class logit $y_1$ is larger than those of all other classes $y_i$. We define $\boldsymbol{z} := [z_2, .., z_n]^T$, skipping the margin of $y_1$ to itself. Under the above assumptions, the statistics of the classification margin for a single classifier are:

$$\mathbb{E}[z_i] = c_1 - c_i$$
$$\mathrm{Var}[z_i] = \sigma_{p,1}^2 + \sigma_{p,i}^2 + \sigma_{c,1}^2 + \sigma_{c,i}^2 - 2\rho_{p,1i}\sigma_{p,1}\sigma_{p,i} - 2\rho_{c,1i}\sigma_{c,1}\sigma_{c,i}$$

**Ensemble** Now, we construct an ensemble of $k$ of these classifiers. Using soft-voting (cf. Eq. (2)) to compute the ensemble output $\bar{\boldsymbol{y}} = \frac{1}{k}\sum_{l=1}^k \boldsymbol{y}^l$. We assume the $\boldsymbol{y}_p^i$ and $\boldsymbol{y}_p^j$ to be correlated with $\zeta_p\Sigma_p$ for classifiers $i \neq j$ and similarly model the correlation of $\boldsymbol{y}_c^l$ with $\zeta_c\Sigma_c$ for $\zeta_p, \zeta_c \in [0,1]$. Letting $\boldsymbol{y}^* := [\boldsymbol{y}^{1\top}, ..., \boldsymbol{y}^{k\top}]^\top$ denote the concatenation of the logit vectors of all classifiers, we assemble their joint covariance matrix in a block-wise manner from the classifier individual covariance matrices as:

$$\mathrm{Cov}[\boldsymbol{y}^*] = \Sigma^* = \begin{bmatrix} \Sigma_p + \Sigma_c & \cdots & \zeta_p\Sigma_p + \zeta_c\Sigma_c \\ & \ddots & \vdots \\ & & \Sigma_p + \Sigma_c \end{bmatrix}$$

We then write the corresponding classification margins $\bar{z}_i = \bar{y}_1 - \bar{y}_i$. or in vector notation $\bar{\boldsymbol{z}} := [\bar{z}_2, .., \bar{z}_m]^T$, again skipping the margin of $\bar{y}_1$ to itself. By linearity of expectation we obtain $\mathbb{E}[\bar{z}_i] = \mathbb{E}[z_i] = c_1 - c_i$ or equivalently

$$\mathbb{E}[\bar{\boldsymbol{z}}] = \bar{\mu} = \begin{pmatrix} c_1 \\ \vdots \\ c_1 \end{pmatrix} - \mathbf{c}_{[2:m]}.$$

We define the ensemble difference matrix $D \in \mathbb{R}^{m-1 \times mk}$ with elements $d_{i,j}$ such that $\bar{z} = Dy^*$:

$$D_{ij} = \begin{cases} \frac{1}{k}, & \text{if } j \bmod m = 1 \\ \frac{-1}{k}, & \text{else if } j \bmod m = i \bmod m, \quad j \in [1, ..., mk], i \in [2, ..., m]. \\ 0, & \text{else} \end{cases}$$

This allows us to write the covariance matrix of the ensemble classification margins

$$\text{Cov}[\bar{z}] = \bar{\Sigma} = D\Sigma^* D^\top.$$

Now we can evaluate its diagonal elements or use the multinomial theorem and rule on the variance of correlated sums to obtain the variance of individual terms:

$$\text{Var}[\bar{z}_i] = \underbrace{\frac{k + 2\binom{k}{2}\zeta_p}{k^2}(\sigma_{p,1}^2 + \sigma_{p,i}^2 - 2\rho_{p,1i}\sigma_{p,1}\sigma_{p,i})}_{\sigma_p^2(k)} + \underbrace{\frac{k + 2\binom{k}{2}\zeta_c}{k^2}(\sigma_{c,1}^2 + \sigma_{c,i}^2 - 2\rho_{c,1i}\sigma_{c,1}\sigma_{c,i})}_{\sigma_c^2(k)}.$$

**Variance Reduction** We can split $\text{Var}[\bar{z}_i]$ into the components associated with the perturbation effect $\sigma_p^2(k)$ and the clean prediction $\sigma_c^2(k)$, all as functions of the ensemble element number $k$.

Now, we analyze these variance components independently by normalizing them with the corresponding components of an individual classifier:

$$\frac{\sigma_p^2(k)}{\sigma_p^2(1)} = \frac{(1 + \zeta_p(k-1))(\sigma_{p,1}^2 + \sigma_{p,i}^2 - 2\rho_{p,1i}\sigma_{p,1}\sigma_{p,i})}{k(\sigma_{p,1}^2 + \sigma_{p,i}^2 - 2\rho_{p,1i}\sigma_{p,1}\sigma_{p,i})} = \frac{1 + \zeta_p(k-1)}{k} \xrightarrow{k \to \infty} \zeta_p$$

$$\frac{\sigma_c^2(k)}{\sigma_c^2(1)} = \frac{(1 + \zeta_c(k-1))(\sigma_{c,1}^2 + \sigma_{c,i}^2 - 2\rho_{c,1i}\sigma_{c,1}\sigma_{c,i})}{k(\sigma_{c,1}^2 + \sigma_{c,i}^2 - 2\rho_{c,1i}\sigma_{c,1}\sigma_{c,i})} = \frac{1 + \zeta_c(k-1)}{k} \xrightarrow{k \to \infty} \zeta_c$$

We observe that both variance components go towards their corresponding correlation coefficients $\zeta_p$ and $\zeta_c$ as ensemble size grows, highlighting the importance of non-identical classifiers.

Especially for samples that are near a decision boundary, this variance reduction will lead to much more consistent predictions, in turn significantly increasing the lower confidence bound $\underline{p}_1$ and thereby the certified radius as per Theorem 3.1.

## D    THEORY VALIDATION

In this section, we present experiments validating the modeling assumptions made in §5 to derive the reduced variance over perturbations in RS resulting from using ensembles of similar classifiers. There are three main assumptions we make:

- The (pre-softmax) output of a single model can be modeled as $f^l(x) = y_p + y_c$, where $y_c$ captures the prediction on clean samples and $y_p$ the effect of perturbations with mean zero.

- The clean and perturbation components $y_c$ and $y_p$ are independent.

- The covariance of $y_p^l$ and $y_c^l$ between different classifiers $i \neq j$ can be parametrized as $\text{Cov}(y_p^i, y_p^j) = \zeta_p \Sigma_p$ and similarly $\text{Cov}(y_c^i, y_c^j) = \zeta_c \Sigma_c$.

To show that these assumptions are valid, we additionally assume multivariate Gaussian distributions and evaluate an ensemble of $k = 50$ `ResNet20` on 100 clean samples $x$, each perturbed with $n = 1000$ different error terms $\epsilon$, and compare the obtained distributions and covariance matrices with our modeling. We generally observe excellent agreement between observation and model at low noise level $\sigma_\epsilon = 0.25$ and good agreement at a high noise level of $\sigma_\epsilon = 1.0$.

**Individual classifier predictions** We model the predictions of individual classifiers $f: \mathbb{R}^d \mapsto \mathbb{R}^m$ on perturbed inputs $x + \epsilon$ for a *single* arbitrary but fixed $x$ and Gaussian perturbations $\epsilon \sim \mathcal{N}(0, \sigma_\epsilon^2 I)$ with $f^l(x) = y_p + y_c$.

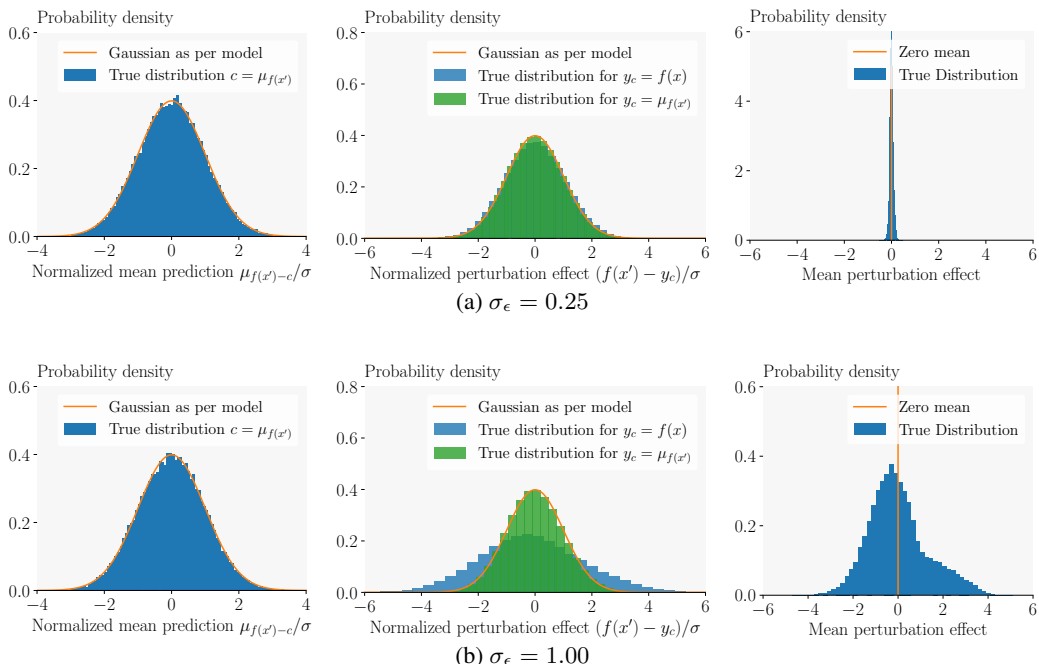

Figure 9: Comparison of modeled and true distributions of clean prediction (left), perturbation effect (middle), and mean perturbation effect (right) for $\sigma_\epsilon = 0.25$ (top) and $\sigma_\epsilon = 0.5$ (bottom).

We can interpret the clean component $\boldsymbol{y}_c$ in two different ways: either as the prediction $\boldsymbol{y}_c := \boldsymbol{f}(\boldsymbol{x})$ on the clean sample $\boldsymbol{x}$ or as the mean prediction over perturbations of this sample $\boldsymbol{y}_c := \mathbb{E}_{\epsilon \sim \mathcal{N}(0, \sigma_\epsilon^2 \boldsymbol{I})}[\boldsymbol{f}(\boldsymbol{x} + \epsilon)]$. In both cases, we assume $\boldsymbol{y}_c$ to be distributed with mean $\boldsymbol{c} \in \mathbb{R}^m$ and covariance $\Sigma_c$. The first case is illustrated in Fig. 8, where we rescale both our model and the predictions with the sample-wise mean and variance over classifiers and observe excellent agreement between model and observed distribution. The second case is illustrated in the leftmost column in Fig. 9 for two different $\sigma_\epsilon$, where we again observe excellent agreement between the true distribution and our model. Here we denote with $\mu_{f(x')}$ the mean prediction over perturbed samples $\mathbb{E}_{\epsilon \sim \mathcal{N}(0, \sigma_\epsilon^2 \boldsymbol{I})}[\boldsymbol{f}(\boldsymbol{x} + \epsilon)]$, corresponding to the setting of $\boldsymbol{y}_c := \mathbb{E}_{\epsilon \sim \mathcal{N}(0, \sigma_\epsilon^2 \boldsymbol{I})}[\boldsymbol{f}(\boldsymbol{x} + \epsilon)]$ and similarly with denote with $\mu_{f(x)}$ the mean prediction over classifiers for an unperturbed sample $\mathbb{E}_l[\boldsymbol{f}^l(\boldsymbol{x})]$, corresponding to the setting of $\boldsymbol{y}_c := \boldsymbol{f}(\boldsymbol{x})$.

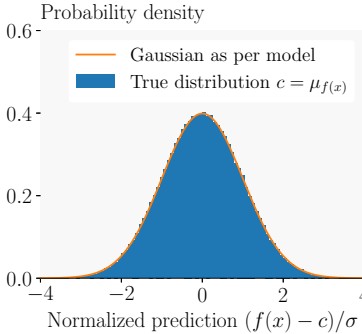

Figure 8: Comparison of the true and modelled distribution of clean predictions $\boldsymbol{y}_c = \boldsymbol{f}(\boldsymbol{x})$.

The perturbation effect $\boldsymbol{y}_p$ is assumed to be mean zero, either following from local linearization and zero mean perturbations when defining the clean component as $\boldsymbol{y}_c := \boldsymbol{f}(\boldsymbol{x})$ or directly from the definition of the clean component as $\boldsymbol{y}_c := \mathbb{E}_{\epsilon \sim \mathcal{N}(0, \sigma_\epsilon)}[\boldsymbol{f}(\boldsymbol{x} + \epsilon)]$. For small perturbations ($\sigma_\epsilon = 0.25$), the assumption of local linearity holds, and we observe a mean perturbation effect distributed very tightly around 0 (see top right pane in Fig. 9). For larger perturbations ($\sigma_\epsilon = 1.00$), this assumption of local linearity begins to break down, and we observe a much wider spread of mean perturbation effect (see bottom right pane in Fig. 9). At both perturbation levels, we observe an excellent agreement of the perturbation effect model with the observed data when using the definition of $\boldsymbol{y}_c := \mathbb{E}_{\epsilon \sim \mathcal{N}(0, \sigma_\epsilon)}[\boldsymbol{f}(\boldsymbol{x} + \epsilon)]$ (green in the middle column in Fig. 9). Using $\boldsymbol{y}_c := \boldsymbol{f}(\boldsymbol{x})$, we still observe great agreement at the lower perturbation magnitude but a notable disagreement at high perturbation levels. While both definitions for $\boldsymbol{y}_c$ are fully compatible with our derivation in §5, we choose to present a version based on $\boldsymbol{y}_c := \boldsymbol{f}(\boldsymbol{x})$ as it allows for a better intuitive understanding.

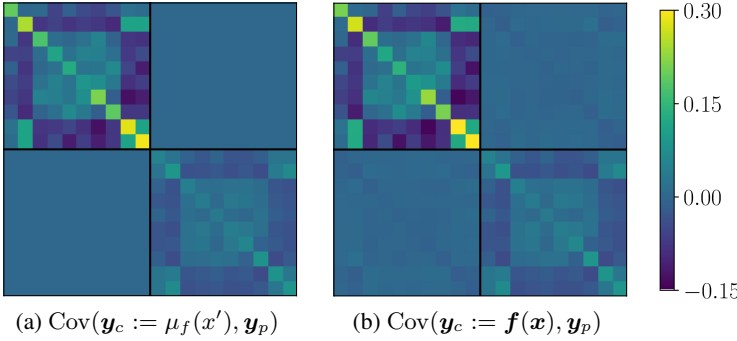

(a) $\mathrm{Cov}(\boldsymbol{y}_c := \mu_f(x'), \boldsymbol{y}_p)$      (b) $\mathrm{Cov}(\boldsymbol{y}_c := \boldsymbol{f}(\boldsymbol{x}), \boldsymbol{y}_p)$

Figure 10: Covariance $\mathrm{Cov}(\boldsymbol{y}_c, \boldsymbol{y}_p)$ of clean and perturbation components $\boldsymbol{y}_c$ and $\boldsymbol{y}_p$, respectively, for $\sigma_\epsilon = 0.25$ using $\boldsymbol{y}_c := \mu_f(x')$ (left) or $\boldsymbol{y}_c := \boldsymbol{f}(\boldsymbol{x})$ (right). The upper diagonal block corresponds to $\mathrm{Cov}(\boldsymbol{y}_c, \boldsymbol{y}_c)$ and the lower one to $\mathrm{Cov}(\boldsymbol{y}_p, \boldsymbol{y}_p)$'.

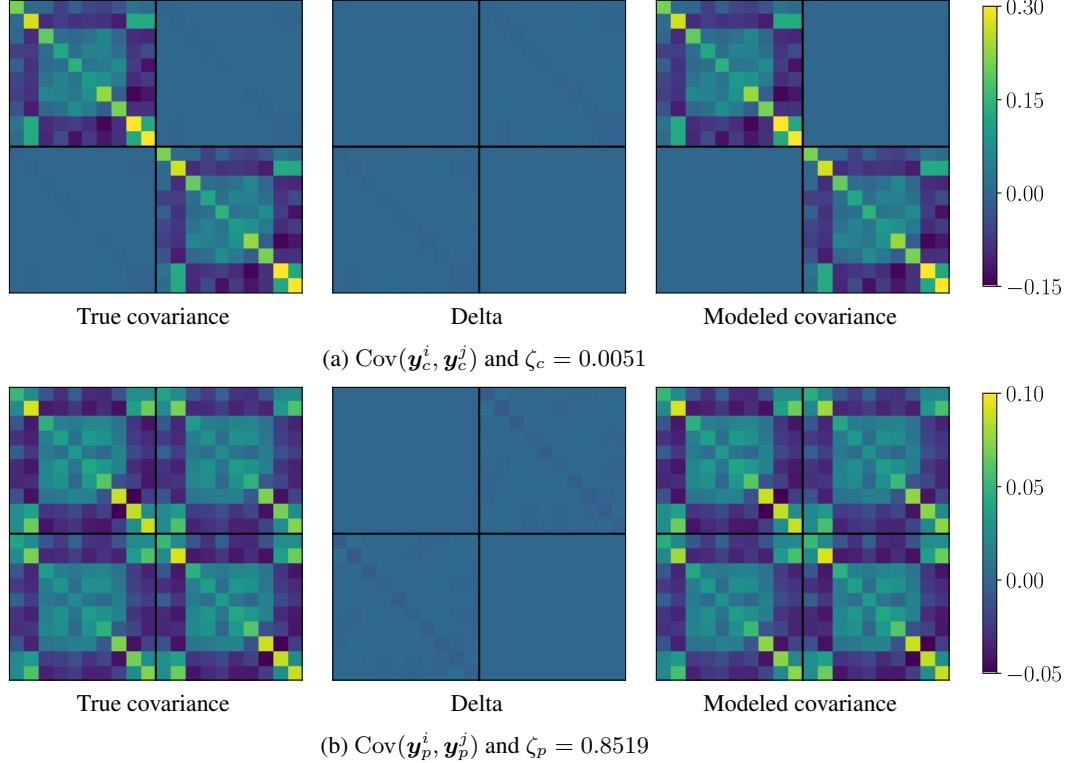

True covariance      Delta      Modeled covariance

(a) $\mathrm{Cov}(\boldsymbol{y}_c^i, \boldsymbol{y}_c^j)$ and $\zeta_c = 0.0051$

True covariance      Delta      Modeled covariance

(b) $\mathrm{Cov}(\boldsymbol{y}_p^i, \boldsymbol{y}_p^j)$ and $\zeta_p = 0.8519$

Figure 11: Comparison of modeled (right) and true (left) covariance matrix between the clean and perturbation components $\boldsymbol{y}_c^l$ (top) and $\boldsymbol{y}_p^l$ (bottom), respectively, of different classifiers for $\sigma_\epsilon = 0.25$ using $\boldsymbol{y}_c := \boldsymbol{f}(\boldsymbol{x})$.

**Correlation between clean and error component**     We model $\boldsymbol{y}_c$ and $\boldsymbol{y}_p$ separately to distinguish between randomness over perturbations, which are introduced during RS, and over the models introduced during training with different random seeds. As $\boldsymbol{y}_c$ models the large scale effects of training and $\boldsymbol{y}_p$ the smaller local scale local effects of inference time perturbations, we assume them to be independent, corresponding to a covariance $\mathrm{Cov}(y_{p,i}, y_{c,j}) \approx 0 \ \forall i, j$. Depending on which definition of $\boldsymbol{y}_c$ we use, we obtain either of the two covariance matrices in Fig. 10, where the upper left-hand block is the covariance of $\boldsymbol{y}_c$, the lower righthand block the covariance of $\boldsymbol{y}_p$ and the off-diagonal blocks $\mathrm{Cov}(y_{p,i}, y_{c,j})$. In the setting of $\boldsymbol{y}_c := \mu_f(x')$, the two components $\boldsymbol{y}_c$ and $\boldsymbol{y}_p$ are perfectly independent as can be seen by the 0 off-diagonal blocks in the covariance matrix shown in Fig. 10a. In the setting of $\boldsymbol{y}_c := \boldsymbol{f}(\boldsymbol{x})$, very small terms can be seen in the off-diagonal blocks shown in Fig. 10b. Overall, the agreement with our model is excellent in both cases.

**Inter classifier correlation structure**    In our analysis, we consider ensembles of similar classifiers which are based on the same architecture and trained using the same methods, just using different random seeds. Hence, we assume that the correlation between the logits of different classifiers has a similar structure but smaller magnitude than the correlation between the logits of the same classifier. Correspondingly, we parametrize the covariance between $\boldsymbol{y}_p^i$ and $\boldsymbol{y}_p^j$ for classifiers $i \neq j$ with $\zeta_p \Sigma_p$ and similarly between $\boldsymbol{y}_c^i$ and $\boldsymbol{y}_c^j$ with $\zeta_c \Sigma_c$ for $\zeta_p, \zeta_c \in [0, 1]$. To evaluate this assumption, we consider all $\binom{50}{2}$ pairings of classifiers and compute the resulting pairwise covariance matrices for the clean components $\boldsymbol{y}_c^l := \boldsymbol{f}^l(\boldsymbol{x})$ and perturbation effects $\boldsymbol{y}_p^l$. In Fig. 11 we show the resulting covariance matrices (left column), our model (right column), and the delta between the two (center column). Again we observe excellent agreement to the point that even given the delta maps, it is hard to spot the differences in the covariance matrices.

**Model fit conclusion**    We conclude that our model is able to capture the relevant behavior even of deep networks (here `ResNet20`) with great accuracy that goes beyond what is required for a theoretical analysis to allow for interesting insights into the underlying mechanics.

# E    PROOF OF THEOREM 6.1

We first restate the theorem:

**Theorem** (Theorem 6.1 restated). *For $\alpha, \beta \in [0, 1], s \in \mathbb{N}^+, n_1 < \cdots < n_s$, CERTIFYADP:*

1. *returns $\hat{c}_A$ if at least $1 - \alpha$ confident that $G$ is robust with a radius of at least $r$.*

2. *returns $\oslash$ before stage $s$ only if at least $1 - \beta$ confident that robustness of $G$ at radius $r$ can not be shown.*

3. *for $n_s \geq \lceil n(1 - \log_\alpha(s)) \rceil$ has maximum certifiable radii at least as large as CERTIFY for $n$.*

*Proof.* We show statements 1, 2, and 3 individually.

*Proof of 1.* If a phase in CERTIFYADP returns $\hat{c}_A$, then via Theorem 3.1, $G$ will be not robust with probability at most $\alpha/s$. Via Bonferroni correction (Bonferroni, 1936), if CERTIFYADP returns $\hat{c}_A$, then $G$ is robust with radius at least $r$ with confidence $1 - s(\alpha/s) = 1 - \alpha$.

*Proof of 2.* If CERTIFYADP returns $\oslash$ in phase $s$, we have $\overline{p_A} < p_A' = \Phi(r/\sigma_\epsilon)$ the minimum success probability for certification at r. By the definition of the UPPERCONFIDENCEBOUND the true success probability of $G$ will be $p_A \leq \overline{p_A}$ with confidence at least $\beta/(s-1)$. Hence, if phase $j$ returns $\oslash$, robustness of $G$ cannot be shown at radius $r$ with confidence $1 - \beta/(s-1)$, even with access to the true success probability or infinitely many samples. Again with Bonferroni correction (Bonferroni, 1936), the overall probability that $G$ is robust at $r$ despite CERTIFYADP abstaining early is at most $(s-1)(\beta/(s-1)) = \beta$.

*Proof of 3.* Finally, to prove the last part, we assume `cnts[`$\hat{c}_A$`] = n`, yielding the largest certifiable radii for any given $n$ and $\alpha$. Now, the largest radius provable via Theorem 3.1 with $n$ samples at $\alpha$ is $\alpha^{1/n}$ (Cohen et al., 2019). Similarly, for $n_s$ samples at $\alpha/s$, it is $\left(\frac{\alpha}{s}\right)^{\frac{1}{n_s}}$. Then we have the following equivalences:

$$(\alpha)^{\frac{1}{n}} = \left(\frac{\alpha}{s}\right)^{\frac{1}{n_s}} \Leftrightarrow \alpha^{n_s} = \left(\frac{\alpha}{s}\right)^n \Leftrightarrow n_s \log \alpha = n(\log \alpha - \log s) \Leftrightarrow n_s = n(1 - n \log_\alpha(s))$$

Hence, if we choose $n_s = \lceil n(1 - \log_\alpha(s)) \rceil$, then we can certify at least the same maximum radius with $n_s$ samples at $\alpha/s$ as with $n$ samples at $\alpha$. Thus, overall, we can certify the same maximum radius at $\alpha$. □

# F    ADDITIONAL RANDOMIZED SMOOTHING DETAILS

This section contains definitions needed for the standard certification via Randomized Smoothing, CERTIFY, proposed by Cohen et al. (2019).

SAMPLEUNDERNOISE$(F, x, n, \sigma_\epsilon)$ first samples $n$ inputs $x_1, \ldots, x_n$ with $x_1 = x + \epsilon_1, \ldots, x_n = x + \epsilon_n$ for $\epsilon_1, \ldots, \epsilon_n \sim \mathcal{N}(0, \sigma_\epsilon)$. Then it counts how often $F$ predicts which class for these $x_1, \ldots, x_n$ and returns the corresponding $m$ dimensional array.

UPPERCONFBND$(k, n, 1 - \alpha)$ returns an upper bound on $p$ with confidence at least $1 - \alpha$ where $p$ is an unknown probability such that $k \sim \mathcal{B}(n, p)$. Here, $\mathcal{B}(n, p)$ is a binomial distribution with parameters $n$ and $p$.

LOWERCONFBND$(k, n, 1 - \alpha)$ is analogous to UPPERCONFBND but returns a lower bound on $p$ instead of an upper bound with confidence $1 - \alpha$.

We point the interested reader to Cohen et al. (2019) for more details about Randomized Smoothing.

## G  EXPERIMENTAL DETAILS

In this section, we provide greater detail on the datasets, architectures, and training and evaluation methods used for our experiments.

### G.1  DATASET DETAILS

We use the MNIST, CIFAR10, and ImageNet datasets in our experiments.

MNIST (Deng, 2012) contains 60'000 training and 10'000 test set images partitioned into 10 classes for the 10 digits. We evaluate all images from the test set.

CIFAR10 (Krizhevsky et al., 2009) (MIT License) contains 50'000 training and 10'000 test set images partitioned into 10 classes. We evaluate every $20^{th}$ image of the test set starting with index 0 (0, 20, 40, ..., 9980) in our experiments, following previous work (Cohen et al., 2019).

ImageNet (Russakovsky et al., 2015) contains 1'287'167 training and 50'000 validation images, partitioned into 1000 classes. We evaluate every $100^{th}$ image of the validation set starting with index 0 (0, 100, 200, ..., 49900) in our experiments, following previous work (Cohen et al., 2019).

### G.2  ARCHITECTURE DETAILS

We use different versions of ResNet (He et al., 2016) for our experiments. Concretely, we evaluate ensembles of ResNet20 and ResNet110 on CIFAR10 and ensembles of ResNet50 on ImageNet.

ResNet110 has about 6.35 times as many parameters as ResNet20 (see Table 6). ResNet50 instantiated for ImageNet has substantially more parameters than ResNet110 instantiated for CIFAR10 because of the significantly larger input dimension of ImageNet samples.

Table 6: Parameter count of the used network architectures

| Dataset | Architecture | Parameter count |
|---------|-------------|-----------------|
| CIFAR10 | ResNet20 | 272'474 |
|         | ResNet110 | 1'730'714 |
| ImageNet | ResNet50 | 25'557'032 |

### G.3  TRAINING METHODS

To obtain high certified radii via RS, the base model $F$ must be trained to cope with the added Gaussian noise $\epsilon$. To this end, Cohen et al. (2019) propose data augmentation with Gaussian noise during training, referred to as GAUSSIAN in the following. Building on this, Salman et al. (2019) suggest SMOOTHADV, a combination of adversarial training (Madry et al., 2018; Kurakin et al., 2017; Rony et al., 2019) with the data augmentation from GAUSSIAN. (Here, we always consider the PGD version.) While improving accuracy, this training procedure is computationally very expensive. MACER (Zhai et al., 2020) achieves similar performance with a cheaper training procedure, adding a loss term directly optimizing a surrogate certification radius. Jeong & Shin (2020), called CONSISTENCY in the following, replace this term with a more easily optimizable loss, further decreasing training time and improving performance. Depending on the setting, the current state-of-the-art results are either achieved by SMOOTHADV, MACER, CONSISTENCY, or a combination thereof.

Further, Salman et al. (2020) present denoised smoothing, where the base classifier $f = h \circ f'$ is the composition of a denoiser $h$ that removes the Gaussian noise from the input and an underlying classifier $f'$ that is not specially adapted to Gaussian noise, e.g., accessed via an API.

### G.4 TRAINING DETAILS

All models are implemented in PyTorch (Paszke et al., 2019) (customized BSD license).

For GAUSSIAN training (No license specified), we use the published code by Cohen et al. (2019). For each network, we chose a different random seed, otherwise using identical parameters. Overall, we trained 50 `ResNet20` and 10 `ResNet110` for each $\sigma_\epsilon \in \{0.25, 0.5, 1.0\}$ on CIFAR10. For ImageNet, we trained 3 `ResNet50` for $\sigma_\epsilon = 1.0$, and took the results for $\sigma_\epsilon \in \{0.25, 0.50\}$ from their GitHub.

For CONSISTENCY training (MIT License), we use the code published by Jeong & Shin (2020) and CONSISTENCY instantiation built on GAUSSIAN training. Similarly to GAUSSIAN, we use a different random seed and otherwise identical parameters for all networks. We chose the parameters reported to yield the largest ACR by Jeong & Shin (2020) for any given $\sigma_\epsilon$, except for $\sigma_\epsilon = 0.25$ on ImageNet, for which no parameters were reported. In detail, for CIFAR10 we generally use $\eta = 0.5$, using $\lambda = 20$ for $\sigma_\epsilon = 0.25$ and $\lambda = 10$ for $\sigma_\epsilon = 0.5$ and $\sigma_\epsilon = 1.0$ (all for both `ResNet20` and `ResNet110`). In this way, we train 50 `ResNet20` and 10 `ResNet110` for each $\sigma_\epsilon \in \{0.25, 0.5, 1.0\}$. For ImageNet, we use $\eta = 0.1$ and $\lambda = 5$ for $\sigma_\epsilon = 1.0$. For $\sigma_\epsilon = 0.25$ and $\sigma_\epsilon = 0.5$, we use $\eta = 0.5$ with $\lambda = 10$ respectively $\lambda = 5$.

For SMOOTHADV training (MIT License), we use the PGD based instantiation of the code published by Salman et al. (2019). Note that due to the long training times, we only train SMOOTHADV models ourselves on CIFAR10. The individual models we train only differ by random seed, all using PGD attacks. For $\sigma_\epsilon = 0.25$, we train with $T = 10$ steps, an $\epsilon = 255/255$, 10 epochs of warm-up, and $m_{\text{train}} = 8$ noise terms per sample during training. For $\sigma_\epsilon = 0.5$, we train with $T = 2$ steps, an $\epsilon = 512/255$, 10 epochs of warm-up, and $m_{\text{train}} = 2$ noise terms per sample during training. Finally, $\sigma_\epsilon = 1.0$, we train with $T = 10$ steps, an $\epsilon = 512/255$, 10 epochs of warm-up, and $m_{\text{train}} = 2$ noise terms per sample during training.

We use the same training schedule and optimizer for all models, i.e., stochastic gradient descent with Nesterov momentum (weight = 0.9, no dampening), with an $\ell_2$ weight decay of $0.0001$. For CIFAR10, we use a batch size of 256 and an initial learning rate of $0.1$, reducing it by a factor of 10 every 50 epochs and training for a total of 150 epochs. For ImageNet, we use the same settings, only reducing the total epoch number to 90 and decreasing the learning rate every 30 epochs.

All single MACER (No license specified) trained models for CIFAR10 are taken directly from Zhai et al. (2020).

For our experiments on denoised smoothing, we use the 4 white box denoisers with DNCNN-WIDE architecture trained with learning schedules 1, 3, 4, and 5 for STAB `ResNet110`, and the `ResNet110` trained on unperturbed samples for 90 epochs from Salman et al. (2020) (MIT License).

To rank the single models for CIFAR10, we have evaluated them on a disjunct hold-out portion of the CIFAR10 test set. Concretely, we use the test images with indices $1, 21, 41, ..., 9981$ to rank the single models for GAUSSIAN, CONSISTENCY, and SMOOTHADV trained models. The performances of individual models values we report are those of the models with the best score on this validation set for CIFAR10 and the best score on the test set for ImageNet (favoring individual models in this setting). For ensembles of size $k < 10$ and $k < 50$ for `ResNet110` and `ResNet20`, respectively, we ensemble the $k$ models according to their performance on the hold-out set. We note that other combinations might yield stronger ensembles, but an exhaustive search of all combinatorially many possibilities is computationally infeasible.

CIFAR10 models were trained on single GeForce RTX 2080 Ti and ImageNet models on quadruple 2080 Tis. We report the epoch-wise and total training times for individual models in Table 7 (when trained sequentially one at a time).

Table 7: Reference training times for individual models on GeForce RTX2080 Tis

| Dataset | Training | Architecture | #GPUs | time per Epoch [s] | total time [h] |
|---|---|---|---|---|---|
| CIFAR10 | GAUSSIAN | ResNet110 | 1 | 41.8 | 1.74 |
| | | ResNet20 | 1 | 9.6 | 0.40 |
| | CONSISTENCY | ResNet110 | 1 | 79.5 | 3.31 |
| | | ResNet20 | 1 | 18.5 | 0.77 |
| | SMOOTHADV ($\sigma_\epsilon = 0.25$) | ResNet110 | 1 | 2471 | 102.96 |
| | SMOOTHADV ($\sigma_\epsilon \in \{0.5, 1.0\}$) | ResNet110 | 1 | 660 | 27.5 |
| ImageNet | CONSISTENCY | ResNet50 | 4 | 3420 | 85 |

### G.5 EXPERIMENT TIMINGS

To evaluate the ensembles of `ResNet20` and `ResNet110` on CIFAR10, we use single GeForce RTX 2080 Ti, and to evaluate ensembles of `ResNet50` on ImageNet, we use double 2080 Tis.

For both datasets, 500 samples are evaluated per experiment as discussed in App. G.1. Below we list the time required for certification using CERTIFY and no $K$-Consensus aggregation:

- `ResNet110` on CIFAR10 (1 GPU): 4.12h per single model; 41.2h per ensemble of 10
- `ResNet20` on CIFAR10 (1 GPU): 0.825h per single model; 41.3h per ensemble of 50
- `ResNet50` on ImageNet (2 GPUs): 8.75h per single model; 26.2h per ensemble of 3

The timing for different ensemble sizes scales linearly between full size and individual model timings. The time required for certification using CERTIFYADP or $K$-Consensus aggregation or both can be obtained by dividing the timings reported above with the speed-up factor TimeRF reported for the corresponding experiments.

## H ADDITIONAL EXPERIMENTS

In the following, we present numerous additional experiments and more detailed results for some of the experiments presented in §7.

### H.1 ADDITIONAL RESULTS ON CIFAR10

In this section, we present more experimental results on CIFAR10. Concretely, Table 8 presents the performance of ensembles of `ResNet20` ($k \in \{1, 5, 50\}$) and `ResNet110` ($k \in \{1, 10\}$) trained using a wide range of of methods for $\sigma_\epsilon = 0.25$, $\sigma_\epsilon = 0.5$, and $\sigma_\epsilon = 1.0$. We again consistently observe that ensembles significantly outperform their constituting models, implying that ensembles are effective for various training methods and model architectures, as well as different $\sigma_\epsilon$. In particular, they lead to a new state-of-the-art both in ACR and at most radii at $\sigma_\epsilon = 0.25$, $\sigma_\epsilon = 0.5$, and $\sigma_\epsilon = 1.0$.

Fig. 12 visualizes the evolution of ACR and certified accuracy with ensemble size for $\sigma_\epsilon = 0.25$. In Fig. 13, we visualize the evolution of certified accuracy over radii for GAUSSIAN trained models.

### H.2 ADDITIONAL RESULTS ON IMAGENET

Extending the results from §7, Table 9 also provides ensemble results on ImageNet for $\sigma_\epsilon = 0.25$, and $\sigma_\epsilon = 0.5$ with CONSISTENCY trained `ResNet50`. Similarly to CIFAR10, we consistently observe that ensembles significantly outperform their constituting models, implying that ensembles are effective on various datasets. In particular, we achieve a new state-of-the-art both in ACR and certified accuracy at most radii for $\sigma_\epsilon = 0.25$, $\sigma_\epsilon = 0.50$, and $\sigma_\epsilon = 1.00$.

Additionally, we report the performance of all individual classifiers which constitute the ensembles, where $k = 3$ combines all three and $k = 2$ the first two for GAUSSIAN and CONSISTENCY, and the last two for SMOOTHADV. Note that the SMOOTHADV models are taken directly from Salman et al. (2019) and were trained with a different $\epsilon$ parameter (256, 512, and 1024). We observe that if there is a large discrepancy between model performance at some radii (e.g., the SMOOTHADV models at

Table 8: Average certified radius (ACR) and certified accuracy at various radii for ensembles of $k$ models ($k = 1$ are single models) for a various training methods and model architectures on CIFAR10. Larger is better.

| $\sigma_\epsilon$ | Training | Architecture | $k$ | ACR | Radius r |||||||||||
|---|---|---|---|---|---|---|---|---|---|---|---|---|---|---|---|
| | | | | | 0.0 | 0.25 | 0.50 | 0.75 | 1.00 | 1.25 | 1.50 | 1.75 | 2.00 | 2.25 | 2.50 |
| 0.25 | GAUSSIAN | ResNet110 | 1 | 0.450 | 77.6 | 60.0 | 45.6 | 30.6 | 0.0 | 0.0 | 0.0 | 0.0 | 0.0 | 0.0 | 0.0 |
| | | | 10 | 0.541 | **83.4** | **70.6** | 55.4 | 42.0 | 0.0 | 0.0 | 0.0 | 0.0 | 0.0 | 0.0 | 0.0 |
| | | ResNet20 | 1 | 0.434 | 77.4 | 63.4 | 43.6 | 26.6 | 0.0 | 0.0 | 0.0 | 0.0 | 0.0 | 0.0 | 0.0 |
| | | | 5 | 0.500 | 80.2 | 65.4 | 50.4 | 36.8 | 0.0 | 0.0 | 0.0 | 0.0 | 0.0 | 0.0 | 0.0 |
| | | | 50 | 0.517 | 80.4 | 68.6 | 52.8 | 38.8 | 0.0 | 0.0 | 0.0 | 0.0 | 0.0 | 0.0 | 0.0 |
| | CONSISTENCY | ResNet110 | 1 | 0.546 | 75.6 | 65.8 | 57.2 | 46.4 | 0.0 | 0.0 | 0.0 | 0.0 | 0.0 | 0.0 | 0.0 |
| | | | 10 | **0.583** | 76.8 | 70.4 | **60.4** | 51.6 | 0.0 | 0.0 | 0.0 | 0.0 | 0.0 | 0.0 | 0.0 |
| | | ResNet20 | 1 | 0.528 | 71.8 | 64.2 | 55.6 | 45.2 | 0.0 | 0.0 | 0.0 | 0.0 | 0.0 | 0.0 | 0.0 |
| | | | 5 | 0.547 | 73.0 | 65.4 | 57.6 | 47.6 | 0.0 | 0.0 | 0.0 | 0.0 | 0.0 | 0.0 | 0.0 |
| | | | 50 | 0.551 | 73.0 | 64.8 | 57.0 | 50.2 | 0.0 | 0.0 | 0.0 | 0.0 | 0.0 | 0.0 | 0.0 |
| | SMOOTHADV | ResNet110 | 1 | 0.527 | 70.4 | 62.8 | 54.2 | 48.0 | 0.0 | 0.0 | 0.0 | 0.0 | 0.0 | 0.0 | 0.0 |
| | | | 10 | 0.560 | 71.6 | 64.8 | 57.8 | **52.4** | 0.0 | 0.0 | 0.0 | 0.0 | 0.0 | 0.0 | 0.0 |
| | MACER | ResNet110 | 1 | 0.518 | 77.4 | 69.0 | 52.6 | 39.4 | 0.0 | 0.0 | 0.0 | 0.0 | 0.0 | 0.0 | 0.0 |
| 0.50 | GAUSSIAN | ResNet110 | 1 | 0.535 | 65.8 | 54.2 | 42.2 | 32.4 | 22.0 | 14.8 | 10.8 | 6.6 | 0.0 | 0.0 | 0.0 |
| | | | 10 | 0.648 | **69.0** | **60.4** | **49.8** | 40.0 | 29.8 | 19.8 | 15.0 | 9.6 | 0.0 | 0.0 | 0.0 |
| | | ResNet20 | 1 | 0.534 | 65.2 | 55.0 | 43.0 | 33.0 | 22.4 | 16.2 | 9.6 | 5.0 | 0.0 | 0.0 | 0.0 |
| | | | 5 | 0.615 | 67.6 | 58.4 | 47.4 | 38.8 | 27.4 | 19.8 | 13.2 | 7.0 | 0.0 | 0.0 | 0.0 |
| | | | 50 | 0.630 | 67.2 | 59.4 | 48.6 | 39.2 | 29.0 | 21.6 | 14.6 | 8.2 | 0.0 | 0.0 | 0.0 |
| | CONSISTENCY | ResNet110 | 1 | 0.708 | 63.2 | 54.8 | 48.8 | 42.0 | 36.0 | 29.8 | 22.4 | 16.4 | 0.0 | 0.0 | 0.0 |
| | | | 10 | **0.756** | 65.0 | 59.0 | 49.4 | **44.8** | 38.6 | 32.0 | 26.2 | 19.8 | 0.0 | 0.0 | 0.0 |
| | | ResNet20 | 1 | 0.691 | 62.6 | 55.2 | 47.4 | 41.8 | 34.6 | 28.4 | 21.8 | 16.8 | 0.0 | 0.0 | 0.0 |
| | | | 5 | 0.723 | 62.2 | 55.0 | 48.6 | 42.6 | 36.4 | 29.8 | 23.4 | 20.6 | 0.0 | 0.0 | 0.0 |
| | | | 50 | 0.729 | 61.6 | 55.8 | 49.2 | 43.0 | 37.8 | 30.6 | 24.2 | 20.0 | 0.0 | 0.0 | 0.0 |
| | SMOOTHADV | ResNet110 | 1 | 0.707 | 52.6 | 47.6 | 46.0 | 41.2 | 37.2 | 31.8 | 28.0 | 23.4 | 0.0 | 0.0 | 0.0 |
| | | | 10 | 0.730 | 52.4 | 48.6 | 45.8 | 42.6 | **38.8** | **34.4** | **30.4** | **25.0** | 0.0 | 0.0 | 0.0 |
| | MACER | ResNet110 | 1 | 0.668 | 62.4 | 54.4 | 48.2 | 40.2 | 33.2 | 26.8 | 19.8 | 13.0 | 0.0 | 0.0 | 0.0 |
| 1.00 | GAUSSIAN | ResNet110 | 1 | 0.532 | 48.0 | 40.0 | 34.4 | 26.6 | 22.0 | 17.2 | 13.8 | 11.0 | 9.0 | 5.8 | 4.2 |
| | | | 10 | 0.607 | 49.4 | **44.0** | 37.6 | 29.6 | 24.8 | 20.0 | 16.4 | 13.6 | 11.2 | 9.4 | 6.8 |
| | | ResNet20 | 1 | 0.538 | 48.0 | 41.2 | 35.0 | 27.8 | 21.6 | 17.8 | 14.8 | 12.0 | 9.0 | 5.6 | 3.4 |
| | | | 5 | 0.590 | 49.2 | 42.8 | 37.8 | 30.4 | 24.0 | 19.4 | 16.2 | 13.8 | 11.2 | 8.2 | 5.0 |
| | | | 50 | 0.597 | **49.6** | 43.0 | 37.4 | 30.4 | 23.6 | 18.6 | 15.8 | 13.6 | 11.2 | 9.0 | 5.0 |
| | CONSISTENCY | ResNet110 | 1 | 0.778 | 45.4 | 41.6 | 37.4 | 33.6 | 28.0 | 25.6 | 23.4 | 19.6 | 17.4 | 16.2 | 14.6 |
| | | | 10 | 0.809 | 46.4 | 42.6 | 37.2 | 33.0 | 29.4 | 25.6 | 23.2 | 21.0 | 17.6 | 16.2 | 14.6 |
| | | ResNet20 | 1 | 0.757 | 43.6 | 39.8 | 34.8 | 30.8 | 27.6 | 24.6 | 22.6 | 19.4 | 17.4 | 15.4 | 13.8 |
| | | | 5 | 0.779 | 43.4 | 40.0 | 35.6 | 32.2 | 28.0 | 24.8 | 22.2 | 20.4 | 17.4 | 15.8 | 13.8 |
| | | | 50 | 0.788 | 45.2 | 40.6 | 36.4 | 32.4 | 28.2 | 24.6 | 22.0 | 20.2 | 17.8 | 16.0 | 14.6 |
| | SMOOTHADV | ResNet110 | 1 | 0.844 | 45.4 | 41.0 | 38.0 | 34.8 | 32.2 | 28.4 | 25.0 | **22.4** | 19.4 | **16.6** | **14.8** |
| | | ResNet110 | 10 | **0.855** | 44.8 | 40.6 | **38.2** | **35.6** | **32.6** | **29.2** | **25.8** | 22.0 | **19.8** | 15.8 | **14.8** |
| | MACER | ResNet110 | 1 | 0.797 | 42.8 | 40.6 | 37.4 | 34.4 | 31.0 | 28.0 | 25.0 | 21.4 | 18.4 | 15.0 | 13.8 |

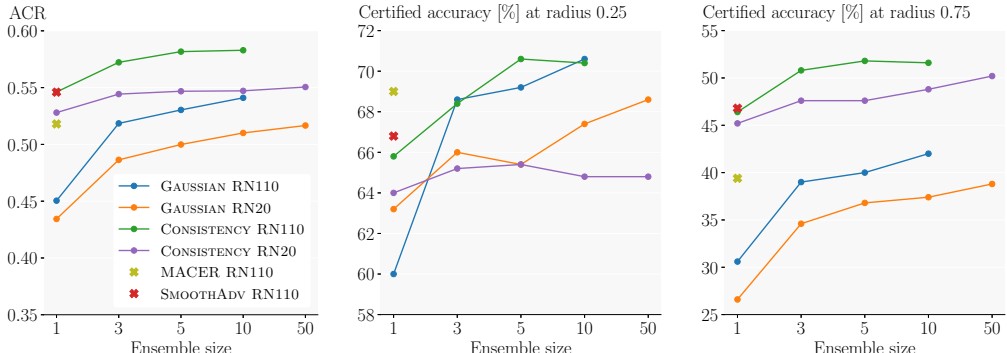

Figure 12: Evolution of average certified radius (ACR), certified accuracy at $r = 0.25$, and $r = 0.75$ with ensemble size $k$ for various underlying models and $\sigma_\epsilon = 0.25$ on CIFAR10.

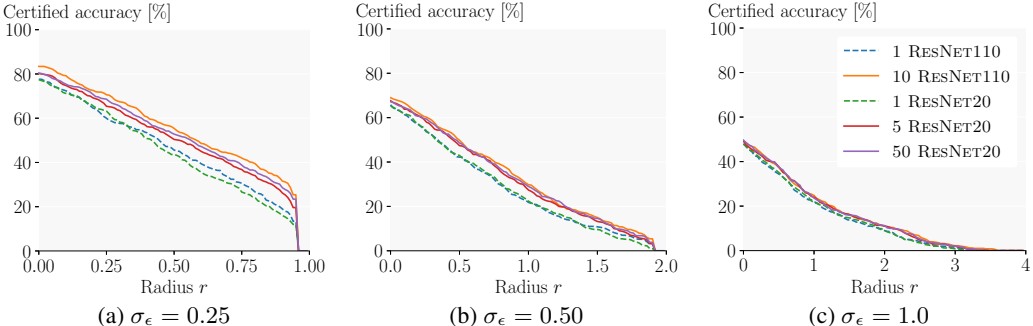

Figure 13: CIFAR10 certified accuracy over certified radius $r$ for GAUSSIAN trained models.

small radii), the ensemble might perform worse than the strongest constituting model. Overall and when performance is homogenous, however, ensembles once again obtain significant performance increases over individual models.

## H.3 ADDITIONAL ABLATION STUDY OF ENSEMBLES

This section presents a range of experiments on different different aspects of variance reduction through ensembles for Randomized Smoothing. In particular, we cover aggregation approaches, the effect of ensemble size, the variability of our results, certifications with an equal number of inferences, ensembles of differently trained models, and ensembles of denoisers.

### H.3.1 AGGREGATION APPROACHES

We experiment with various instantiations of the general aggregation approach described in §4. In particular, we consider soft-voting, hard-voting, soft-voting after softmax, and weighted soft-voting.

**Soft-voting**  We simply average the logits to obtain

$$\bar{f}(\boldsymbol{x}) = \frac{1}{k} \sum_{l=1}^{k} f^l(\boldsymbol{x}).$$

**Hard-voting**  We process the outputs of single models with the post-processing function $\gamma_{HV}(\boldsymbol{y}^l) = \mathbf{1}_{j=\arg\max_i(y_i^l)}$ that provides a one-hot encoding of the $\arg\max$ of $f^l(\boldsymbol{x})$ before averaging to obtain

$$\bar{f}(\boldsymbol{x}) = \frac{1}{k} \sum_{l=1}^{k} \mathbf{1}_{j=\arg\max_i(f^l(\boldsymbol{x})_i)}.$$

**Soft-voting after softmax**  We process the outputs of the single models by applying the softmax function as post-processing function $\gamma_{softmax}$ to obtain

$$\bar{f}(\boldsymbol{x}) = \frac{1}{k} \sum_{l=1}^{k} \frac{\exp(f^l(\boldsymbol{x}))}{\sum_i \exp(f^l(\boldsymbol{x})_i)}.$$

**Weighted soft-voting**  We consider soft-voting with a classifier-wise weights $w^l$ learned on a separate holdout set and obtain

$$\bar{f}(\boldsymbol{x}) = \frac{1}{k} \sum_{l=1}^{k} w^l f^l(\boldsymbol{x}).$$

We compare these approaches in Table 10 and observe that the two soft-voting schemes perform very similarly and outperform the other approaches. We decide to use soft-voting without soft-max for its conceptual simplicity for all other experiments.

Table 9: Average certified radius (ACR) and certified accuracy at various radii for ensembles of k models (k = 1 are single models) for a various training methods on ImageNet. Larger is better.

| $\sigma_\epsilon$ | Training | $k$ | ACR | Radius r 0.0 | 0.50 | 1.00 | 1.50 | 2.00 | 2.50 | 3.00 | 3.50 |
|---|---|---|---|---|---|---|---|---|---|---|---|
| | GAUSSIAN | 1 | 0.477 | 66.7 | 49.4 | 0.0 | 0.0 | 0.0 | 0.0 | 0.0 | 0.0 |
| | | 1 | 0.512 | 63.0 | 54.0 | 0.0 | 0.0 | 0.0 | 0.0 | 0.0 | 0.0 |
| | | 1 | 0.516 | 64.8 | 54.2 | 0.0 | 0.0 | 0.0 | 0.0 | 0.0 | 0.0 |
| 0.25 | CONSISTENCY | 1 | 0.509 | 64.4 | 53.8 | 0.0 | 0.0 | 0.0 | 0.0 | 0.0 | 0.0 |
| | | 2 | 0.538 | 65.0 | 56.2 | 0.0 | 0.0 | 0.0 | 0.0 | 0.0 | 0.0 |
| | | 3 | **0.545** | 65.6 | **57.0** | 0.0 | 0.0 | 0.0 | 0.0 | 0.0 | 0.0 |
| | SMOOTHADV [†] | 1 | 0.528 | 65 | 56 | 0 | 0 | 0 | 0 | 0 | - |
| | MACER [†] | 1 | 0.544 | **68** | **57** | 0 | 0 | 0 | 0 | 0 | - |
| | GAUSSIAN | 1 | 0.733 | 57.2 | 45.8 | 37.2 | 28.6 | 0.0 | 0.0 | 0.0 | 0.0 |
| | | 1 | 0.793 | 56.0 | 48.0 | 39.6 | 34.0 | 0.0 | 0.0 | 0.0 | 0.0 |
| | | 1 | 0.806 | 55.4 | 48.8 | 42.2 | 35.0 | 0.0 | 0.0 | 0.0 | 0.0 |
| 0.5 | CONSISTENCY | 1 | 0.799 | 54.0 | 48.0 | 41.2 | 35.2 | 0.0 | 0.0 | 0.0 | 0.0 |
| | | 2 | 0.851 | 58.6 | 51.6 | 43.6 | 37.0 | 0.0 | 0.0 | 0.0 | 0.0 |
| | | 3 | **0.868** | 57.0 | 52.0 | **44.6** | **38.4** | 0.0 | 0.0 | 0.0 | 0.0 |
| | SMOOTHADV [†] | 1 | 0.815 | 54 | 49 | 43 | 37 | 0 | 0 | 0 | - |
| | MACER [†] | 1 | 0.831 | **64** | **53** | 43 | 31 | 0 | 0 | 0 | - |
| | | 1 | 0.849 | 42.4 | 35.6 | 30.0 | 25.2 | 20.2 | 15.4 | 12.2 | 10.0 |
| | | 1 | 0.856 | 42.4 | 37.2 | 31.4 | 25.8 | 19.6 | 15.4 | 12.0 | 8.4 |
| | GAUSSIAN | 1 | 0.839 | 40.8 | 35.6 | 29.4 | 25.8 | 20.4 | 15.6 | 12.2 | 8.0 |
| | | 2 | 0.944 | 45.0 | 37.6 | 33.4 | 29.8 | 22.8 | 18.0 | 14.6 | 11.0 |
| | | 3 | 0.968 | 43.8 | 38.4 | 34.4 | 29.8 | 23.2 | 18.2 | 15.4 | 11.4 |
| | | 1 | 1.022 | 43.2 | 39.8 | 35.0 | 29.4 | 24.4 | 22.2 | 16.6 | 13.4 |
| | | 1 | 0.990 | 42.0 | 37.2 | 34.4 | 29.6 | 24.8 | 20.2 | 16.0 | 13.4 |
| 1.0 | CONSISTENCY | 1 | 1.006 | 41.6 | 38.6 | 35.2 | 29.6 | 25.4 | 21.2 | 17.6 | 13.8 |
| | | 2 | 1.086 | 44.8 | 41.0 | 36.6 | 32.4 | 27.4 | 22.4 | 19.4 | 15.6 |
| | | 3 | **1.108** | 44.6 | 40.2 | **37.2** | **34.0** | **28.6** | 23.2 | 20.2 | 16.4 |
| | | 1 | 1.011 | 40.6 | 38.6 | 33.8 | 29.8 | 25.6 | 20.6 | 18.0 | 14.4 |
| | | 1 | 1.002 | 39.4 | 35.4 | 32.0 | 29.2 | 25.6 | 22.2 | 20.0 | 16.4 |
| | SMOOTHADV | 1 | 0.927 | 32.0 | 30.8 | 28.6 | 26.0 | 23.6 | 21.0 | 19.2 | 18.6 |
| | | 2 | 1.022 | 37.4 | 33.4 | 31.4 | 29.4 | 26.6 | 23.8 | 21.0 | 18.6 |
| | | 3 | 1.065 | 38.6 | 36.0 | 34.0 | 30.0 | 27.6 | **24.6** | **21.2** | **18.8** |
| | MACER [†] | 1 | 1.008 | **48** | **43** | 36 | 30 | 25 | 18 | 14 | - |

[†] As reported by Zhai et al. (2020).

Table 10: Comparison of various aggregation methods for ensembles of 10 `ResNet110` at $\sigma_\epsilon = 0.25$ on CIFAR10. Larger is better.

| Training | Aggregation method | ACR | Radius r | | | |
|---|---|---|---|---|---|---|
| | | | 0.00 | 0.25 | 0.50 | 0.75 |
| GAUSSIAN | soft-voting | **0.541** | 83.4 | 70.6 | **55.4** | **42.0** |
| | hard-voting | 0.529 | 83.8 | 69.4 | 53.2 | 40.6 |
| | soft-voting after softmax | 0.540 | **84.0** | **70.8** | 55.2 | 41.8 |
| | weighted soft-voting | 0.538 | 83.4 | 70.4 | 54.6 | 41.8 |
| CONSISTENCY | soft-voting | 0.583 | 76.8 | 70.4 | 60.4 | 51.6 |
| | hard-voting | 0.574 | **77.2** | 69.6 | 60.0 | 50.6 |
| | soft-voting after softmax | **0.584** | 77.0 | 70.4 | **60.6** | **51.8** |
| | weigthed soft-voting | 0.579 | 76.4 | **70.6** | 59.6 | 51.2 |

Table 11: Effect of ensemble size $k$ on ACR and certified accuracy at different radii, for GAUSSIAN and CONSISTENCY trained `ResNet20`. Larger is better.

| Training | $\sigma_\epsilon$ | $k$ | ACR | Radius r | | | | | | | | | |
|---|---|---|---|---|---|---|---|---|---|---|---|---|---|
| | | | | 0.0 | 0.25 | 0.50 | 0.75 | 1.00 | 1.25 | 1.50 | 1.75 | 2.00 | 2.25 | 2.50 |
| GAUSSIAN | 0.25 | 1 | 0.434 | 77.4 | 63.4 | 43.6 | 26.6 | 0.0 | 0.0 | 0.0 | 0.0 | 0.0 | 0.0 | 0.0 |
| | | 3 | 0.486 | 79.6 | 66.0 | 49.6 | 34.6 | 0.0 | 0.0 | 0.0 | 0.0 | 0.0 | 0.0 | 0.0 |
| | | 5 | 0.500 | 80.2 | 65.4 | 50.4 | 36.8 | 0.0 | 0.0 | 0.0 | 0.0 | 0.0 | 0.0 | 0.0 |
| | | 10 | 0.510 | **80.4** | 67.4 | 51.8 | 37.4 | 0.0 | 0.0 | 0.0 | 0.0 | 0.0 | 0.0 | 0.0 |
| | | 50 | **0.517** | **80.4** | **68.6** | **52.8** | **38.8** | 0.0 | 0.0 | 0.0 | 0.0 | 0.0 | 0.0 | 0.0 |
| | 1.0 | 1 | 0.538 | 48.0 | 41.2 | 35.0 | 27.8 | 21.6 | 17.8 | 14.8 | 12.0 | 9.0 | 5.6 | 3.4 |
| | | 3 | 0.579 | 49.2 | 42.8 | 36.4 | 30.2 | 23.0 | 18.8 | 15.6 | 13.0 | 10.8 | 7.4 | 4.4 |
| | | 5 | 0.590 | 49.2 | 42.8 | **37.8** | 30.4 | 24.0 | 19.4 | 16.2 | 13.8 | 11.2 | 8.2 | **5.0** |
| | | 10 | 0.592 | 48.8 | 42.8 | 37.0 | 30.4 | 23.6 | 19.0 | 16.2 | 13.6 | 11.0 | **9.2** | **5.0** |
| | | 50 | **0.597** | 49.6 | **43.0** | 37.4 | 30.4 | 23.6 | 18.6 | 15.8 | 13.6 | **11.2** | 9.0 | **5.0** |
| CONSISTENCY | 0.25 | 1 | 0.528 | 71.8 | 64.2 | 55.6 | 45.2 | 0.0 | 0.0 | 0.0 | 0.0 | 0.0 | 0.0 | 0.0 |
| | | 3 | 0.544 | **73.2** | 65.2 | **57.6** | 47.6 | 0.0 | 0.0 | 0.0 | 0.0 | 0.0 | 0.0 | 0.0 |
| | | 5 | 0.547 | 73.0 | **65.4** | **57.6** | 47.6 | 0.0 | 0.0 | 0.0 | 0.0 | 0.0 | 0.0 | 0.0 |
| | | 10 | 0.547 | 72.2 | 65.0 | 57.4 | 48.8 | 0.0 | 0.0 | 0.0 | 0.0 | 0.0 | 0.0 | 0.0 |
| | | 50 | **0.551** | 73.0 | 64.8 | 57.0 | **50.2** | 0.0 | 0.0 | 0.0 | 0.0 | 0.0 | 0.0 | 0.0 |
| | 1.0 | 1 | 0.757 | 43.6 | 39.8 | 34.8 | 30.8 | 27.6 | 24.6 | **22.6** | 19.4 | 17.4 | 15.4 | 13.8 |
| | | 3 | 0.777 | 44.2 | 39.6 | **36.8** | 31.8 | 28.4 | **25.0** | 22.4 | 19.8 | 17.6 | 15.6 | 14.0 |
| | | 5 | 0.779 | 43.4 | 40.0 | 35.6 | 32.2 | 28.0 | 24.8 | 22.2 | **20.4** | 17.4 | 15.8 | 13.8 |
| | | 10 | 0.784 | 44.4 | **40.6** | 36.4 | 32.2 | **29.2** | 24.2 | 22.2 | **20.4** | 17.6 | **16.0** | 14.2 |
| | | 50 | **0.788** | **45.2** | **40.6** | 36.4 | **32.4** | 28.2 | 24.6 | 22.0 | 20.2 | **17.8** | **16.0** | **14.6** |

### H.3.2 ENSEMBLE SIZE

In Table 11, we present extended results on the effect of ensemble size for different training methods and $\sigma_\epsilon$. Generally, we observe that ensembles of sizes 3 and 5 already significantly improve over the performance of a single model, while further increasing the ensemble size mostly leads to marginal gains. This aligns well with our theory from §5.

### H.3.3 VARIABILITY OF RESULTS

In Table 12, we report mean and standard deviation of ACR and certified accuracy for 50 GAUSSIAN and CONSISTENCY trained `ResNet20` with different random seeds, either combined to 10 ensembles with $k = 5$ or evaluated individually. Generally, we observe a notably smaller standard deviation of the ensembles compared to individual models. We visualize this in Fig. 14, where we show the $\pm 3\,\sigma$ region.

### H.3.4 EQUAL NUMBER OF INFERENCES

In Table 13, we compare the certified accuracy of an ensemble of $k = 10$ `ResNet110` obtained with $n = 100'000$ samples

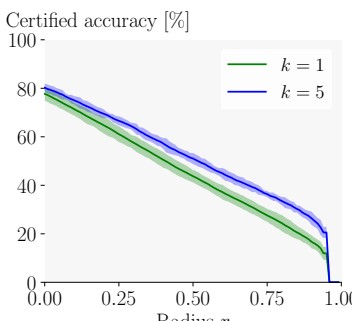

Figure 14: Mean and $\pm 3\,\sigma$ interval of certified accuracy over radius $r$ computed over 50 GAUSSIAN trained `ResNet20`.

Table 12: Mean and standard deviation of ACR and certified accuracy for CIFAR10 at various radii for 50 `ResNet20` with different random seeds either evaluated as 10 ensembles with $k = 5$ or as individual models at $\sigma_\epsilon = 0.25$.

| Training | $k$ | ACR | Radius r | | |
|---|---|---|---|---|---|
| | | | 0.25 | 0.50 | 0.75 |
| GAUSSIAN | 1 | $0.4350 \pm 0.0046$ | $61.08 \pm 1.08$ | $43.74 \pm 0.86$ | $27.58 \pm 0.88$ |
| | 5 | $0.4994 \pm 0.0025$ | $66.58 \pm 0.62$ | $51.04 \pm 0.76$ | $36.38 \pm 0.42$ |
| CONSISTENCY | 1 | $0.5202 \pm 0.0049$ | $63.57 \pm 0.86$ | $54.22 \pm 1.03$ | $44.30 \pm 0.89$ |
| | 5 | $0.5445 \pm 0.0017$ | $64.90 \pm 0.66$ | $56.86 \pm 0.49$ | $47.94 \pm 0.68$ |

Table 13: Comparing and ensemble and an individual model at an equal number of total inferences on CIFAR10: average certified radius (ACR) and certified accuracy at various radii for ensembles of $k$ models ($k = 1$ are single models) for a various sampling sizes $n$. Larger is better.

| Training | $k$ | $n$ | ACR | Radius r | | | | | | | | |
|---|---|---|---|---|---|---|---|---|---|---|---|---|
| | | | | 0.0 | 0.25 | 0.50 | 0.75 | 1.00 | 1.25 | 1.50 | 1.75 | 2.00 |
| GAUSSIAN | 1 | 100'000 | 0.535 | 65.8 | 54.2 | 42.2 | 32.4 | 22.0 | 14.8 | 10.8 | 6.6 | 0.0 |
| | 1 | 1'000'000 | 0.552 | 65.8 | 54.6 | 42.2 | 32.6 | 22.4 | 15.2 | 11.0 | 7.0 | **3.8** |
| | 10 | 100'000 | **0.648** | **69.0** | **60.4** | **49.8** | **40.0** | **29.8** | **19.8** | **15.0** | 9.6 | 0.0 |
| CONSISTENCY | 1 | 100'000 | 0.708 | 63.2 | 54.8 | 48.8 | 42.0 | 36.0 | 29.8 | 22.4 | 16.4 | 0.0 |
| | 1 | 1'000'000 | 0.752 | 63.2 | 54.8 | 48.8 | 42.0 | 36.4 | 30.4 | 23.0 | 17.8 | **14.6** |
| | 10 | 100'000 | **0.756** | **65.0** | **59.0** | **49.4** | **44.8** | **38.6** | **32.0** | **26.2** | **19.8** | 0.0 |

on CIFAR10 with that of an individual model obtained with $n = 1'000'000$, i.e. at the same computational cost.

For both GAUSSIAN and CONSISTENCY training, we observe that at all radii which can mathematically be certified using $n = 100'000$ samples the ensemble it significantly outperforms the individual model at equal computational cost. Note that the individual models with $n = 1'000'000$ samples have the same or just a marginally larger certified accuracy at all radii $\leq 1.75$ than the same individual model with $n = 100'000$ samples. Only at radius 2.00 where certification is mathematically impossible with $n = 100'000$ samples does using additional samples increase the certified accuracy. Note however, that, in contrast to our ensembles, this does not influence the actual accuracy of the underlying model in any way, but simply allows to derive tighter confidence bounds. Table 14 shows similar results for ImageNet and ensembles of $k = 3$ `ResNet110`, implying that these observations hold for various datasets.

### H.3.5 ENSEMBLES OF DIFFERENTLY TRAINED MODELS

In Table 15 we show how ensembles built from differently trained base models (SMOOTHADV, MACER, and CONSISTENCY) perform. We observe that the ensemble of all three models is strictly better than any individual model, suggesting that ensembles also work for models trained with different training methods. However, the improvements are significantly smaller than the improvements of ensembling similar classifiers in the same setting (see also Table 8 for comparison). This is again due to a very heterogeneous model performance at some radii, where the strongest models are weighed down by weaker ones.

### H.3.6 ENSEMBLES OF DENOISERS

It was shown (Salman et al., 2020) that standard neural networks can be robustified with respect to Gaussian noise by training a suitable denoiser and evaluating the original model on the denoised sample. Here, we aim to overcome the drawback of any ensemble, the need to train multiple diverse models, by instead training different denoisers and building an ensemble by combining these denoisers with a single underlying model. We show the general applicability of our ensembling approach by

Table 16: Denoised smoothing ensembles on CIFAR10 at $\sigma_\epsilon = 0.25$

| $k$ | ACR | Radius r | | | |
|---|---|---|---|---|---|
| | | 0.0 | 0.25 | 0.50 | 0.75 |
| 1 | 0.378 | **76.4** | 56.6 | 36.8 | 19.2 |
| 4 | **0.445** | 75.2 | **62.2** | **45.8** | **29.0** |

Table 14: Comparing and ensemble and an individual model at an equal number of total inferences on ImageNet: average certified radius (ACR) and certified accuracy at various radii for ensembles of $k$ models ($k = 1$ are single models) for a various sampling sizes $n$. Larger is better.

| Training | $k$ | $n$ | ACR | Radius r | | | | | | | | |
|---|---|---|---|---|---|---|---|---|---|---|---|---|
| | | | | 0.0 | 0.50 | 1.00 | 1.50 | 2.00 | 2.50 | 3.00 | 3.50 | 4.00 |
| | 1 | 100'000 | 1.022 | 43.2 | 39.8 | 35.0 | 29.4 | 24.4 | 22.2 | 16.6 | 13.4 | 0.0 |
| CONSISTENCY | 1 | 300'000 | 1.060 | 43.2 | 39.6 | 35.2 | 29.6 | 24.6 | 22.4 | 16.8 | 14.2 | **11.4** |
| | 3 | 100'000 | **1.108** | **44.6** | **40.2** | **37.2** | **34.0** | **28.6** | **23.2** | **20.2** | **16.4** | 0.0 |

Table 15: Building ensembles with differently trained base models: average certified radius (ACR) and certified accuracy at various radii for ensembles and their constituting models. Larger is better.

| Model | ACR | Radius r | | | |
|---|---|---|---|---|---|
| | | 0.0 | 0.25 | 0.50 | 0.75 |
| SMOOTHADV | 0.542 | 74.4 | 65.4 | 56.2 | 48.0 |
| MACER | 0.518 | 77.8 | 69.0 | 52.6 | 39.0 |
| CONSISTENCY | 0.546 | 75.2 | 66.0 | 57.0 | 46.6 |
| Ensemble (SMOOTHADV and CONSISTENCY) | **0.572** | 75.6 | 68.6 | **59.8** | **50.6** |
| Ensemble (SMOOTHADV, MACER and CONSISTENCY) | 0.567 | **78.4** | **69.4** | 58.8 | 48.0 |

applying it to the setting of denoised smoothing (Salman et al., 2020). We consider 4 pre-trained denoisers and combine them with a single standard `ResNet110` (see App. G.3 for details). Comparing the strongest resulting model with an ensemble of all 4 at $\sigma_\epsilon = 0.25$, we observe a 17% ACR improvement. Training strong denoisers with current methods is challenging. However, based on these results we are confident that advances in denoiser training combined with ensembles can present an efficient way to obtain strong provable models.

### H.3.7 Additional Norms

Below, we demonstrate that our approach generalizes beyond the $\ell_2$-norm setting and generalizes well to improve the robust accuracy against $\ell_0$-, $\ell_1$- and $\ell_\infty$-norm bounded attacks.

$\ell_\infty$-norm robustness is typically derived directly from $\ell_2$-norm bounds (Yang et al., 2020; Salman et al., 2019), and hence our results for the $\ell_2$-norm are directly applicable.

In Table 17 and Table 18, we show the effectiveness of our approach for $\ell_0$-norm bounded perturbations using the training and certification method from Lee et al. (2019). For both, we use the default hyper-parameter $\alpha$ (which is not the same as the $\alpha$ in standard Randomized Smoothing from Cohen et al. (2019)), i.e. $\alpha = 0.8$ for MNIST and $\alpha = 0.2$ for ImageNet. We observe that even an ensemble of just 3 models outperforms the individual models in every setting.

In Table 19, we similarly show the effectiveness of our approach for $\ell_1$-norm bounded perturbations using training with uniform noise and the uniform noise based certification method from Yang et al. (2020) with $\sigma_\epsilon = \lambda = 1.0$. We observe again that even an ensemble of just 3 models outperforms the single model in every setting.

### H.4 Additional Computational Overhead Reduction Experiments

This section contains additional experiments for the computational overhead methods introduced in §6. Whenever we use CERTIFYADP, we set $\beta = 0.001$.

**Adaptive sampling and $K$-consensus** Table 20 is an extensive version of Table 3. In Table 21, we show the effect of adaptive sampling and $K$-Consensus aggregation for GAUSSIAN trained `ResNet110` on CIFAR10, analogously to Table 20. Certifying an ensemble with CERTIFYADP and $K$-Consensus aggregation achieves approximately the same certified accuracy as CERTIFY while the certification time is up to 67 times faster. In particular, for both training methods and for each radius, the certifying the ensemble of $k = 10$ networks with CERTIFYADP and $K$-Consensus aggregation is faster than certifying an individual model with CERTIFY while the certified accuracy is significantly improved

Table 17: Effect of ensemble size $k$ on ACR and certified accuracy at different radii for $l_0$ norm on MNIST. Larger is better.

| k | ACR | Radius r | | | | | | | | | |
|---|---|---|---|---|---|---|---|---|---|---|---|
| | | 0 | 1 | 2 | 3 | 4 | 5 | 6 | 7 | 8 | 9 |
| 1 | 3.231 | 97.8 | 90.0 | 73.7 | 49.9 | 48.5 | 23.6 | 17.3 | 7.9 | 1.6 | 1.6 |
| 3 | 3.575 | **98.4** | **92.5** | 79.0 | 56.1 | 54.7 | 38.2 | 21.6 | 10.7 | **2.4** | **2.4** |
| 10 | **3.612** | **98.4** | **92.5** | **79.1** | **57.0** | **55.6** | **38.9** | **22.1** | **11.1** | **2.4** | **2.4** |

Table 18: Effect of ensemble size k on ACR and certified accuracy at different radii for $l_0$ norm on ImageNet. Larger is better.

| k | ACR | Radius r | | | | | | | | | |
|---|---|---|---|---|---|---|---|---|---|---|---|
| | | 0 | 1 | 2 | 3 | 4 | 5 | 6 | 7 | 8 | 9 |
| 1 | 2.340 | 53.8 | 48.8 | 42.0 | 30.8 | 25.2 | 24.6 | 23.6 | 22.2 | 16.8 | **0.0** |
| 3 | **2.986** | **57.4** | **53.6** | **50.4** | **39.0** | **33.8** | **33.0** | **32.8** | **30.0** | **26.0** | **0.0** |

in most cases. In addition, we confirm our previous observation that the sample reduction is most prominent for small radii (for a given $\sigma_\epsilon$) while KCR increases with radius.

### H.4.1 Additional Adaptive Sampling Experiments

This section contains other experiments with CERTIFYADP, highlighting various aspects of adaptive sampling for ensembles and also individual models.

**Adaptive sampling for ensembles on CIFAR10**  For comparison with Table 3 (respectively Table 20 and Table 21), where we show the combined effect of $K$-Consensus aggregation and adaptive sampling, we show the results for just applying adaptive sampling to the same setting in Table 22, using $\{n_j\} = \{100, 1'000, 10'000, 120'000\}$, $\sigma_\epsilon = 0.25$, and ensembles of 10 CONSISTENCY trained ResNet110. We observe that the speed-ups for small radii are comparable to applying both $K$-Consensus aggregation and adaptive sampling, as only a few samples have to be evaluated. The additional benefit due to $K$-Consensus aggregation grows as the later certification stages are entered more often for larger radii, highlighting the complementary nature of the two methods. Note that for radius $r = 0.75$, 100 samples, and even $1'000$ are never sufficient for certification but can still yield early abstentions.

**Adaptive sampling for ImageNet ensembles**  In Table 23, we demonstrate the applicability of our adaptive sampling approach to ensembles of ResNet50 on ImageNet. In particular, we achieve sampling reduction factors of up to 42 without incurring a significant accuracy penalty. For a fixed $\sigma_\epsilon$, we empirically observe larger speed-ups at smaller radii. This effect, however, depends on the underlying distribution of success probabilities and would be expected to invert for a model that has true success probabilities close to $p_A \approx 0.5$ for most samples.

**Effect of sampling stage sizes**  Additionally, in Table 24, we investigate how the choice of the number of phases and the number of samples in each phase influence the sampling gain for various radii and models. We observe that as few as 100 samples can be sufficient to certify or abstain from up to $80\%$ of samples, making schedules with many phases attractive. However, when the true success probability is close to the one required for certification, many samples are required to obtain sufficiently tight confidence bounds on the success probability to decide either way. This leads to many samples being discarded for schedules with many phases. Additionally, the higher per phase confidence required due to Bonferroni correction requires more samples to be drawn to be able to obtain the same maximum lower confidence bound to the success probability. This latter effect becomes especially pronounced for larger radii, where higher success probabilities are required.

**Ablation of $\alpha$ and $\beta$**  In Fig. 15, we illustrate the effect of different $\alpha$ and $\beta$ on the first phase of certification with CERTIFYADP where we consider just 100 perturbations. Increasing $\alpha$ reduces

Table 19: Effect of ensemble size k on ACR and certified accuracy at different radii for $l_1$ norm on CIFAR10 for models with `ResNet20` architecture. Larger is better.

| k | ACR | Radius r | | | |
|---|---|---|---|---|---|
| | | 0.0 | 0.25 | 0.50 | 0.75 |
| 1 | 0.571 | 71.2 | 65.4 | 60.2 | 51.6 |
| 3 | 0.610 | 72.4 | 68.6 | 63.4 | 56.0 |
| 10 | **0.622** | **72.4** | **69.6** | **64.8** | **57.2** |

Table 20: Adaptive sampling ($\{n_j\} = \{100, 1'000, 10'000, 120'000\}$) and 5-Consensus agg. on CIFAR10 for 10 CONSISTENCY `ResNet110`. SampleRF and TimeRF are the reduction factors in comparison to standard sampling with $n = 100'000$ (larger is better). $ASR_j$ is the % of certification attempts returned in phase $j$. KCR is the % of inputs for which only $K$ classifiers were evaluated.

| Radius | $\sigma_\epsilon$ | $acc_{cert}$ [%] | $ASR_1$ [%] | $ASR_2$ [%] | $ASR_3$ [%] | $ASR_4$ [%] | SampleRF | KCR [%] | TimeRF |
|---|---|---|---|---|---|---|---|---|---|
| 0.25 | 0.25 | 70.4 | 84.8 | 10.2 | 4.2 | 0.8 | **55.24** | 39.3 | **67.16** |
| 0.50 | 0.25 | 60.6 | 20.8 | 69.8 | 6.4 | 3.0 | 18.09 | 57.3 | 25.07 |
| 0.75 | 0.25 | 52.0 | 27.4 | 8.8 | 55.2 | 8.6 | 5.68 | 87.8 | 10.06 |
| 1.00 | 0.50 | 38.6 | 41.2 | 47.0 | 7.8 | 4.0 | 14.79 | 78.3 | 24.01 |
| 1.25 | 0.50 | 32.2 | 46.6 | 9.8 | 37.6 | 6.0 | 8.14 | 92.7 | 15.06 |
| 1.50 | 0.50 | 26.2 | 50.0 | 11.2 | 29.6 | 9.2 | 6.41 | **97.5** | 12.31 |
| 1.75 | 1.00 | 20.8 | 57.6 | 32.0 | 7.4 | 3.0 | 89.1 | 19.02 | 33.14 |
| 2.00 | 1.00 | 17.6 | 61.8 | 27.4 | 8.0 | 2.8 | 91.0 | 19.93 | 35.26 |
| 2.25 | 1.00 | 16.2 | 68.2 | 21.2 | 7.8 | 2.8 | 93.4 | 20.27 | 36.66 |
| 2.50 | 1.00 | 14.6 | 69.6 | 7.8 | 20.6 | 2.0 | 91.4 | 19.38 | 34.45 |

the confidence $1 - \alpha$ required for certification and hence makes early certification more likely for the same underlying true success probabilities (see Fig. 15a). Increasing $\beta$ reduces the confidence $1 - \beta$ that certification will not be possible required for early termination and hence makes early termination more likely for the same underlying true success probabilities (see Fig. 15b). Overall, we observe that evaluating even only 100 perturbations allows us to, with high probability, abstain from or certify samples with a wide range of true success probabilities. We only have to continue with the costly certification process for samples in a narrow band of true success probabilities (see Fig. 15c), greatly reducing the expected certification cost.

### H.4.2 ADDITIONAL K-CONSENSUS EXPERIMENTS

In Table 25, we present a more detailed version of Table 5. We emphasize that for $K = 5$ and $K = 10$ for `ResNet110` and `ResNet20`, respectively, we achieve the same ACR as with the full ensemble while reducing the certification time by a factor of 1.59 and 2.01, respectively.

Table 21: Adaptive sampling with $\{n_j\} = \{100, 1'000, 10'000, 120'000\}$ and $K$-consensus early stopping with $K = 5$ on CIFAR10 with an ensemble of 10 GAUSSIAN trained `ResNet110`. Sample and time gain are the reduction factor in comparison to standard sampling with $n = 100'000$ (larger is better). $ASR_j$ is the percentage of samples returned in phase $j$. KCR is the percentage of samples for which only $K$ classifiers were evaluated.

| Radius | $\sigma_\epsilon$ | acc$_{\text{cert}}$ [%] | $ASR_1$ | $ASR_2$ | $ASR_3$ | $ASR_4$ | SampleRF | KCR [%] | TimeRF |
|--------|-------------------|-------------------------|---------|---------|---------|---------|----------|---------|--------|
| 0.25 | 0.25 | 70.6 | 76.4 | 17.2 | 4.4 | 2.0 | 28.80 | 26.8 | 32.63 |
| 0.50 | 0.25 | 55.4 | 30.4 | 59.2 | 7.8 | 2.6 | 19.80 | 63.5 | 28.73 |
| 0.75 | 0.25 | 42.0 | 39.4 | 10.2 | 41.8 | 8.6 | 6.19 | 89.7 | 11.14 |
| 1.00 | 0.50 | 30.2 | 53.4 | 31.6 | 10.8 | 4.2 | 13.89 | 77.5 | 22.60 |
| 1.25 | 1.00 | 20.2 | 74.4 | 16.4 | 7.0 | 2.2 | 24.93 | 71.4 | 38.31 |
| 1.50 | 1.00 | 16.6 | 66.8 | 26.0 | 5.4 | 1.8 | 29.34 | 75.7 | 45.80 |
| 1.75 | 1.00 | 13.8 | 73.0 | 20.6 | 4.6 | 1.8 | 30.61 | 84.5 | 51.18 |
| 2.00 | 1.00 | 11.2 | 78.4 | 14.6 | 5.4 | 1.6 | **32.97** | 88.5 | **56.80** |
| 2.25 | 1.00 | 9.4 | 82.4 | 8.0 | 6.8 | 2.8 | 21.32 | 92.7 | 38.60 |
| 2.50 | 1.00 | 6.4 | 82.4 | 6.4 | 6.8 | 4.4 | 14.77 | **95.4** | 27.63 |

Table 22: Effect of adaptive sampling with $\{n_j\} = \{100, 1'000, 10'000, 120'000\}$ for ensembles of 10 GAUSSIAN respectively CONSISTENCY trained `ResNet110` on CIFAR10 with $\sigma_\epsilon = 0.25$. $ASA_j$ and $ASC_j$ are the percentages of samples abstained from and certified in phase $j$, respectively

| Radius | Training | acc$_{\text{cert}}$ [%] | $ASA_1$ | $ASC_1$ | $ASA_2$ | $ASC_2$ | $ASA_3$ | $ASC_3$ | $ASA_4$ | $ASC_4$ | SampleRF | TimeRF |
|--------|----------|-------------------------|---------|---------|---------|---------|---------|---------|---------|---------|----------|--------|
| 0.25 | GAUSSIAN | 70.6 | 62.4 | 13.2 | 10.0 | 7.6 | 3.8 | 1.2 | 0.4 | 1.4 | 30.48 | 31.30 |
|  | CONSISTENCY | 70.4 | 73.6 | 10.4 | 7.0 | 4.8 | 3.0 | 0.6 | 0.0 | 0.6 | **66.73** | **67.93** |
| 0.75 | GAUSSIAN | 42.2 | 0.0 | 39.0 | 0.0 | 9.8 | 37.8 | 4.8 | 5.6 | 3.0 | 6.16 | 6.32 |
|  | CONSISTENCY | 52.0 | 0.0 | 27.4 | 0.0 | 8.8 | 51.8 | 3.4 | 4.8 | 3.8 | 5.68 | 5.83 |

Table 23: Adaptive sampling with $\{n_j\} = \{100, 1'000, 10'000, 120'000\}$ on ImageNet with an ensemble of 3 CONSISTENCY trained `ResNet50`. SampleRF and TimeRF are the reduction factor in comparison certification with CERTIFY with $n = 100'000$. $ASR_j$ is the percentage of samples returned in phase $j$.

| Radius | $\sigma_\epsilon$ | acc$_{\text{cert}}$ [%] | $ASR_1$ | $ASR_2$ | $ASR_3$ | $ASR_4$ | SampleRF | TimeRF |
|--------|-------------------|-------------------------|---------|---------|---------|---------|----------|--------|
| 0.50 | 0.25 | 56.8 | 21.2 | 75.0 | 3.0 | 0.8 | **43.00** | **42.32** |
| 1.00 | 0.50 | 44.8 | 31.0 | 59.4 | 7.0 | 2.6 | 20.14 | 20.31 |
| 1.50 | 0.50 | 38.4 | 40.2 | 7.6 | 46.2 | 6.0 | 7.57 | 7.74 |
| 2.00 | 1.00 | 28.6 | 48.6 | 39.4 | 9.2 | 2.8 | 18.98 | 19.19 |
| 2.50 | 1.00 | 23.2 | 55.8 | 8.0 | 31.8 | 4.4 | 10.49 | 10.70 |
| 3.00 | 1.00 | 20.2 | 58.6 | 11.4 | 24.4 | 5.6 | 9.69 | 9.88 |
| 3.50 | 1.00 | 16.4 | 65.4 | 7.4 | 3.2 | 24.0 | 3.12 | 3.20 |

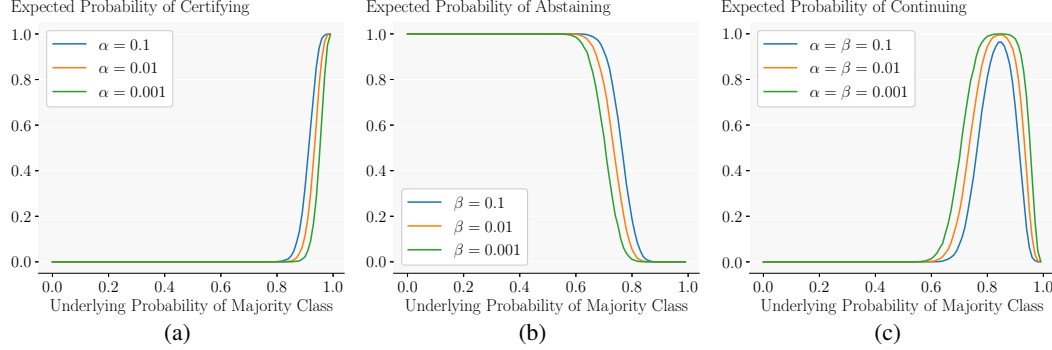

Figure 15: Expected probability of certifying (a), abstaining (b) or continuing to the next phase (c) of CERTIFYADP after the first phase, conditioned on the underlying probability of the majority class. We consider various $\alpha$ and $\beta$. We are in the setting with $r = \sigma_\epsilon = 0.25$ with stage sizes $n_j = \{100, 1'000, 10'000, 120'000\}$, i.e. the first phase whose simulated outcomes we observe here has 100 samples.

Table 24: Effect of adaptive sampling for individual `ResNet110` on CIFAR10 at $\sigma_\epsilon = 0.25$. $\text{ASA}_j$ and $\text{ASC}_j$ are the percentages of samples abstained from and certified in phase $j$, respectively. We compare multiple sampling count progressions $\{n_j\}$.

| Radius | Training | $\{n_j\}$ | $\text{acc}_{cert}$ [%] | $\text{ASA}_1$ | $\text{ASC}_1$ | $\text{ASA}_2$ | $\text{ASC}_2$ | $\text{ASA}_3$ | $\text{ASC}_3$ | $\text{ASA}_4$ | $\text{ASC}_4$ | $\text{ASA}_5$ | $\text{ASC}_5$ | SampleRF | TimeRF |
|---|---|---|---|---|---|---|---|---|---|---|---|---|---|---|---|
| 0.25 | GAUSSIAN | 100, 110'000 | 60.0 | 52.0 | 22.6 | 13.2 | 12.2 | 0.0 | 0.0 | 0.0 | 0.0 | 0.0 | 0.0 | 3.56 | 3.70 |
| | | 1'000, 110'000 | 60.0 | 62.0 | 30.2 | 3.2 | 4.6 | 0.0 | 0.0 | 0.0 | 0.0 | 0.0 | 0.0 | 10.34 | 10.45 |
| | | 10'000, 110'000 | 60.0 | 63.6 | 33.6 | 1.6 | 1.2 | 0.0 | 0.0 | 0.0 | 0.0 | 0.0 | 0.0 | 7.59 | 7.90 |
| | | 50'000, 110'000 | 60.2 | 64.8 | 34.0 | 0.4 | 0.8 | 0.0 | 0.0 | 0.0 | 0.0 | 0.0 | 0.0 | 1.95 | 1.98 |
| | | 100, 1'000, 116'000 | 60.0 | 51.8 | 21.4 | 10.2 | 9.2 | 3.0 | 4.4 | 0.0 | 0.0 | 0.0 | 0.0 | 11.06 | 11.18 |
| | | 100, 10'000, 116'000 | 60.2 | 53.2 | 21.0 | 10.6 | 12.4 | 1.4 | 1.4 | 0.0 | 0.0 | 0.0 | 0.0 | 16.61 | 16.77 |
| | | 1'000, 10'000, 116'000 | 60.0 | 61.4 | 30.2 | 1.8 | 3.2 | 2.0 | 1.4 | 0.0 | 0.0 | 0.0 | 0.0 | 17.01 | 17.21 |
| | | 30'000, 60'000 ,116'000 | 60.0 | 64.4 | 33.6 | 0.4 | 0.6 | 0.4 | 0.6 | 0.0 | 0.0 | 0.0 | 0.0 | 3.08 | 3.21 |
| | | 100, 1'000, 10'000, 120'000 | 60.0 | 53.6 | 21.8 | 7.8 | 8.2 | 2.0 | 3.6 | 1.8 | 1.2 | 0.0 | 0.0 | 20.40 | 21.22 |
| | | 30'000, 60'000, 90'000, 120'000 | 60.0 | 64.6 | 34.0 | 0.6 | 0.0 | 0.0 | 0.2 | 0.0 | 0.6 | 0.0 | 0.0 | 3.09 | 3.22 |
| | | 100, 800, 6'400, 24'000, 123'000 | 60.0 | 53.2 | 21.6 | 8.2 | 7.8 | 2.0 | 3.4 | 1.2 | 0.8 | 0.6 | 1.2 | **24.32** | **24.67** |
| | CONSISTENCY | 100, 110'000 | 66.0 | 68.6 | 16.2 | 8.0 | 7.2 | 0.0 | 0.0 | 0.0 | 0.0 | 0.0 | 0.0 | 5.92 | 6.05 |
| | | 1'000, 110'000 | 66.0 | 75.0 | 20.8 | 1.6 | 2.6 | 0.0 | 0.0 | 0.0 | 0.0 | 0.0 | 0.0 | 17.50 | 17.95 |
| | | 10'000 110'000 | 66.0 | 75.8 | 22.4 | 0.8 | 1.0 | 0.0 | 0.0 | 0.0 | 0.0 | 0.0 | 0.0 | 8.29 | 8.50 |
| | | 50'000, 110'000 | 66.0 | 76.2 | 22.6 | 0.4 | 0.8 | 0.0 | 0.0 | 0.0 | 0.0 | 0.0 | 0.0 | 1.95 | 1.99 |
| | | 100, 1'000, 116'000 | 66.0 | 68.0 | 14.8 | 7.0 | 6.0 | 1.6 | 2.6 | 0.0 | 0.0 | 0.0 | 0.0 | 19.09 | 19.57 |
| | | 100, 10'000, 116'000 | 66.0 | 67.8 | 15.4 | 8.2 | 6.8 | 0.6 | 1.2 | 0.0 | 0.0 | 0.0 | 0.0 | 25.23 | 25.85 |
| | | 1'000, 10'000, 116'000 | 66.0 | 74.8 | 20.6 | 1.2 | 1.6 | 0.6 | 1.2 | 0.0 | 0.0 | 0.0 | 0.0 | 27.44 | 28.02 |
| | | 30'000, 60'000, 116'000 | 66.0 | 76.0 | 22.4 | 0.2 | 0.4 | 0.4 | 0.6 | 0.0 | 0.0 | 0.0 | 0.0 | 3.11 | 3.19 |
| | | 100, 1'000, 10'000, 120'000 | 66.0 | 69.0 | 15.8 | 5.6 | 5.2 | 1.4 | 1.2 | 0.6 | 1.2 | 0.0 | 0.0 | 33.91 | 34.67 |
| | | 30'000, 60'000, 90'000, 120'000 | 66.0 | 76.0 | 22.4 | 0.4 | 0.4 | 0.2 | 0.2 | 0.0 | 0.4 | 0.0 | 0.0 | 3.10 | 3.19 |
| | | 100, 800, 6'400, 24'000, 123'000 | 66.0 | 68.0 | 14.8 | 6.6 | 6.0 | 1.2 | 1.4 | 0.6 | 0.0 | 0.2 | 1.2 | **35.32** | **36.18** |
| 0.75 | GAUSSIAN | 100, 110'000 | 30.6 | 0.0 | 51.2 | 30.8 | 18.0 | 0.0 | 0.0 | 0.0 | 0.0 | 0.0 | 0.0 | 1.86 | 1.88 |
| | | 1'000, 110'000 | 30.4 | 0.0 | 61.0 | 30.6 | 8.4 | 0.0 | 0.0 | 0.0 | 0.0 | 0.0 | 0.0 | 2.27 | 2.34 |
| | | 10'000, 110'000 | 30.4 | 24.6 | 65.4 | 6.0 | 4.0 | 0.0 | 0.0 | 0.0 | 0.0 | 0.0 | 0.0 | 4.74 | 4.89 |
| | | 50'000, 110'000 | 30.4 | 30.2 | 66.8 | 0.4 | 2.6 | 0.0 | 0.0 | 0.0 | 0.0 | 0.0 | 0.0 | 1.87 | 1.93 |
| | | 100, 1'000, 116'000 | 30.4 | 0.0 | 53.8 | 0.0 | 7.4 | 30.6 | 8.2 | 0.0 | 0.0 | 0.0 | 0.0 | 2.19 | 2.25 |
| | | 100, 10'000, 116'000 | 30.8 | 0.0 | 51.6 | 25.0 | 13.4 | 6.0 | 4.0 | 0.0 | 0.0 | 0.0 | 0.0 | 6.02 | 6.02 |
| | | 1'000, 10'000, 116'000 | 30.4 | 0.0 | 60.8 | 25.6 | 3.8 | 5.0 | 4.8 | 0.0 | 0.0 | 0.0 | 0.0 | 6.11 | 6.14 |
| | | 30'000, 60'000, 116'000 | 30.6 | 30.2 | 66.2 | 0.4 | 0.8 | 0.2 | 2.2 | 0.0 | 0.0 | 0.0 | 0.0 | 2.86 | 2.87 |
| | | 100, 1'000, 10'000, 120'000 | 30.2 | 0.0 | 48.8 | 0.0 | 11.6 | 25.6 | 3.8 | 4.8 | 5.4 | 0.0 | 0.0 | 5.92 | 6.10 |
| | | 30'000, 60'000, 90'000, 120'000 | 30.4 | 29.2 | 66.0 | 1.0 | 0.8 | 0.2 | 0.6 | 0.2 | 2.0 | 0.0 | 0.0 | 2.61 | 2.69 |
| | | 100, 800, 6'400, 24'000, 123'000 | 30.2 | 0.0 | 48.8 | 0.0 | 10.8 | 21.2 | 3.6 | 7.2 | 2.8 | 2.0 | 3.6 | **7.24** | **7.49** |
| | CONSISTENCY | 100, 110'000 | 46.0 | 0.0 | 34.6 | 48.8 | 16.6 | 0.0 | 0.0 | 0.0 | 0.0 | 0.0 | 0.0 | 1.39 | 1.42 |
| | | 1'000, 110'000 | 46.2 | 0.0 | 41.2 | 49.0 | 9.8 | 0.0 | 0.0 | 0.0 | 0.0 | 0.0 | 0.0 | 1.52 | 1.55 |
| | | 10'000, 110'000 | 46.2 | 43.8 | 45.8 | 5.2 | 5.2 | 0.0 | 0.0 | 0.0 | 0.0 | 0.0 | 0.0 | 4.65 | 4.70 |
| | | 50'000, 110'000 | 46.4 | 48.6 | 48.2 | 0.6 | 2.6 | 0.0 | 0.0 | 0.0 | 0.0 | 0.0 | 0.0 | 1.87 | 1.89 |
| | | 100, 1'000, 116'000 | 46.4 | 0.0 | 33.8 | 0.0 | 8.8 | 49.2 | 8.2 | 0.0 | 0.0 | 0.0 | 0.0 | 1.48 | 1.50 |
| | | 100, 10'000, 116'000 | 46.2 | 0.0 | 34.4 | 43.6 | 12.2 | 5.4 | 4.4 | 0.0 | 0.0 | 0.0 | 0.0 | 5.52 | 5.61 |
| | | 1'000, 10'000, 116'000 | 47.0 | 0.0 | 41.0 | 44.2 | 5.2 | 5.6 | 4.0 | 0.0 | 0.0 | 0.0 | 0.0 | 5.52 | 5.60 |
| | | 30'000, 60'000, 116'000 | 46.4 | 47.6 | 47.8 | 1.2 | 0.4 | 0.4 | 2.6 | 0.0 | 0.0 | 0.0 | 0.0 | 2.75 | 2.79 |
| | | 100, 1'000, 10'000, 120'000 | 46.4 | 0.0 | 32.4 | 0.0 | 8.2 | 44.2 | 5.2 | 5.0 | 5.0 | 0.0 | 0.0 | 5.32 | 5.40 |
| | | 30'000, 60'000, 90'000, 120'000 | 46.8 | 47.6 | 47.0 | 1.0 | 1.4 | 1.0 | 0.0 | 0.0 | 2.0 | 0.0 | 0.0 | 2.60 | 2.64 |
| | | 100, 800, 6'400, 24'000, 123'000 | 46.4 | 0.0 | 33.6 | 0.0 | 7.2 | 41.0 | 4.4 | 6.0 | 1.4 | 2.2 | 4.2 | **6.37** | **6.47** |

Table 25: Effect of $K$-Consensus aggregation on CONSISTENCY trained ensembles of 10 `ResNet110` and 50 `ResNet20` on CIFAR10 at $\sigma_\epsilon = 0.25$.

| Architecture | $K$ | ACR | Radius r | | | | TimeRF | KCR [%] |
|---|---|---|---|---|---|---|---|---|
| | | | 0.0 | 0.25 | 0.50 | 0.75 | | |
| ResNet110 | 2 | 0.576 | 77.2 | 70.2 | 60.0 | 50.4 | 3.25 | 85.8 |
| | 3 | 0.581 | 77.0 | 70.0 | 60.6 | 51.6 | 2.29 | 79.7 |
| | 5 | 0.583 | 76.8 | 70.4 | 60.4 | 51.6 | 1.59 | 74.2 |
| | 10 | 0.583 | 76.8 | 70.4 | 60.4 | 51.6 | 1.00 | 0.0 |
| ResNet20 | 2 | 0.544 | 72.2 | 65.2 | 57.0 | 48.4 | 6.50 | 87.7 |
| | 3 | 0.549 | 72.6 | 65.0 | 57.4 | 50.0 | 4.41 | 82.4 |
| | 5 | 0.550 | 72.8 | 65.0 | 57.0 | 50.2 | 2.99 | 76.4 |
| | 10 | 0.551 | 73.0 | 64.8 | 57.0 | 50.2 | 2.01 | 69.8 |
| | 50 | 0.551 | 73.0 | 64.8 | 57.0 | 50.2 | 1.00 | 0.0 |

