# OpenReview forum: "Boosting Randomized Smoothing with Variance Reduced Classifiers"
_ICLR.cc/2022/Conference — ICLR 2022 Spotlight_

### Official Review · Reviewer_RVtN · 2021-10-20

**Correctness:** 3
**Technical Novelty And Significance:** 3
**Empirical Novelty And Significance:** 2
**Recommendation:** 8
**Confidence:** 4

**Main Review:**

Strengths

+Important research problem

+Well written and well-organized

+Interesting analysis on why ensemble works

+Extensive evaluation

Weaknesses

-Evaluation only on the L2 certification

-Missing related work

**Summary Of The Paper:**

The paper aims to boost randomized smoothing (RS). Specifically, the paper demonstrates that an ensemble of diverse base models can enhance RS both theoretically and empirically. The key insight is that reducing variance of ensembles over the introduced perturbations can lead to more consistent classifications for inputs. The paper also introduce two simple  yet effective techniques to speed up the certification. Extensive experiments are conducted to thoroughly evaluate the proposed boosted RS.


**Summary Of The Review:**

Certifying an ensemble of models is faster than certifying an individual model through the proposed two strategies. However, training an ensemble of models requires far more computational resources and time than training an individual model. Although the authors mainly leverage the pretrained models, but in practical applications this may be infeasible. I would suggest the authors discuss the issues of training an ensemble of models.

The authors say that “…substantially increased certifiable radii for samples close to the decision boundary..” in abstract, but no results to validate this claim. I would like to see the results for those samples having a small margin.

The authors assume that y_c is the clean part and y_p is the perturbation/noisy part. How to define y_p and y_c when evaluating the real image datasets? Or y_p and y_c are just for analysis?

What’s the impact of $\alpha$ and $\beta$?

In table 1, why ensemble is less effective than MACER when radius r is small?

“Even when using only K = 2, we already obtain 81% and 70% of the ACR improvement obtainable by always evaluating the full ensembles (k = 10 and k = 50) “ => How do you calculate the numbers 81% and 70%?

 In Table 3, what do SampleRF, KRC, and TimeRF mean or short for?

My another concern is that the experimental results are only for certifying L2 perturbation.  There are several papers that design RS to certify $L_0$, $L_1$, $L_\infty$ perturbation.  For example,

$L_0$: Wang et al. “Certified robustness of graph neural networks against adversarial structural perturbation via Randomized Smoothing”

$L_1$: Lee et al., “Tight Certificates of Adversarial Robustness for Randomly Smoothed Classifiers”, NeurIPS 2019

$L_\infty$: Yang et al., Randomized Smoothing of All Shapes and Sizes

I am also really interested in whether the proposed ensemble is also effective for certifying $L_0$, $L_1$, or/and $L_\infty$ perturbation.

The following papers also derive certified robustness based on randomized smoothing:

Jia et al., “Certified Robustness for Top-k Predictions against Adversarial Perturbations via Randomized Smoothing”

Zhang et al., “Black-Box Certification with Randomized Smoothing: A Functional Optimization Based Framework”

Mohapatra et al., “Higher-Order Certification for Randomized Smoothing”

Kumar et al., “Certifying Confidence via Randomized Smoothing”

Kumar et al., “Curse of Dimensionality on Randomized Smoothing for Certifiable Robustness”

Fischer  et al., “Certified Defense to Image Transformations via Randomized Smoothing”

---

> ### Author Response · Authors · 2021-11-15
> **Response to Reviewer RVtN**
>
> We want to thank reviewer $\textcolor{blue}{RVtN}$ for providing insightful feedback, helpful comments, and raising interesting questions, which we answer here:
>
> **Does this approach generalize beyond $\ell_2$-norm bounded perturbations?**
> We follow related work [1,2,3] in using $\ell_2$-norm robustness as a benchmark. However, we now conducted additional experiments and found our approach to also be highly effective against $\ell_0$-, $\ell_1$-, and $\ell_\infty$-norm bounded perturbations using the certification methods from Lee et al. [4] and ​​Yang et al. [5]. We provide detailed results in the newly added Section H.3.7 in the appendix.
>
> **Can you expand your discussion of Randomized Smoothing extensions?**
> Yes, we provide an extensive review in Appendix B to give readers a broader overview of the interesting field of RS. As we showed in the example of different $\ell_p$ norms, many of these extensions are orthogonal to our approach and benefit from our improvements.
>
> **Can you discuss the computational requirements of training multiple models to the ensemble?**
> Yes, we agree that the cost of training ensembles is an important consideration despite our method being able to also reuse (pre-)trained models. Therefore, we are particularly happy to find that with, e.g., 3 or 5 ResNet20 we can outperform an individual ResNet110 (see Section 7.2, ‘Ensemble Size and Model Size Ablation’) while only having 47% and 79% of the parameters and similar total training time (see appendix G.4). This makes RS more accessible as training these smaller models has much lower hardware requirements.
>
> **Why do you expect the increase in certified radius to be particularly pronounced for samples close to the decision boundary?**
> Conceptually, reducing the variance of predictions is only helpful if at least some perturbed samples are misclassified, which we take as a sample being close to the decision boundary. This is illustrated in Figure 2.
> Additionally, we now conducted an experiment to show this more explicitly, where for a large set of inputs:
> - We compute the number of perturbations classified to the majority $n_A$ and runner up class $n_B$, as well as the certification radius for each sample for the individual constituent models.
> - We order the samples by their mean (across classifiers) margin $n_A - n_B$.
> - We compute $n_A$, $n_B$, and the certification radius using the ensemble of these classifiers.
> - We observe that the margin increases 300-times as much for samples in the 25th to 50th percentile (those not directly on but close to a decision boundary), compared to those in the 75th to 100th percentile (those furthest from the decision boundary).
> - This then translates to an almost 4 times as large increase in certified radius for these samples.
>
> **Are $y_c$ and $y_p$ only used for analysis or also during inference? If the latter, how are they computed?**
> Indeed, $y_p$ and $y_c$ are only used for the theoretical analysis of our approach and hence are not computed during inference.
>
> **What is the impact of $\alpha$ and $\beta$ on CertifyAdp?**
> When CertifyAdp outputs a certified radius for a sample, it will be robust to perturbations of this radius with confidence $1-\alpha$ (same as [6]). Additionally, when CertifyAdp abstains early, it is at least $1-\beta$  confident that the given radius can not be certified. Intuitively, by adding $\beta$, we allow for a small chance to abstain even if we could have certified the sample while gaining significant certification speed. We added a paragraph in Section H.4.1 discussing and visualizing these effects in greater detail.
>
> **Why do ensembles yield lower certified accuracy at small radii than the individual MACER model?**
> MACER trained models generally perform particularly well at small radii. Since we had not trained an ensemble of multiple MACER models, the individual MACER model outperforms the ensembles of models trained with different methods at small radii.  In section H.3.5 (Table 15) we show that an ensemble of SmoothAdv, MACER and Consistency trained models outperforms the best individual model in each setting, including at smaller radii.
>
> **How is the relative ACR improvement calculated for K-Consensus aggregation?**
> We compute the ACR increase of using an ensemble with $K$-Consensus aggregation compared to a single model and divide it by the ACR improvement of the full ensemble over a single model, e.g., $\frac{0.576-0.546}{0.583-0.546} = 81\\%$.
>
> **What do KRC, SampleRF, and TimeRF in Table 3 stand for?**
> KCR is the percentage of inputs for which only $K$ classifiers were evaluated ($K$-Consensus aggregation returns early). SampleRF and TimeRF are the factors by which sample complexity and certification time are reduced, respectively. We have moved their definition to the paragraph directly next to Table 3.
>
> We hope to have answered all of the reviewer's questions, are happy to go into more detail regarding any of them and look forward to their response.

---

> > ### Author Response · Authors · 2021-11-15
> > **Response to Reviewer RVtN - References**
> >
> >
> > **References**
> >
> > [1] Jeong and Shin, “Consistency Regularization for Certified Robustness of Smoothed Classifiers”, NeurIPS20
> >
> > [2] Salman et al., “Provably Robust Deep Learning via Adversarially Trained Smoothed Classifiers”, NeurIPS19
> >
> > [3] Zhai et al., “MACER: Attack-free and Scalable Robust Training via Maximizing Certified
> > Radius”, ICLR20
> >
> > [4] Lee et al. “Tight Certificates of Adversarial Robustness for Randomly Smoothed Classifiers”, NeurIPS19
> >
> > [5] ​​Yang et al.,  “Randomized Smoothing of All Shapes and Sizes”, ICML20
> >
> > [6] Cohen et al. “Certified Adversarial Robustness via Randomized Smoothing”, ICML19

---

> > ### Comment · Reviewer_RVtN · 2021-11-18
> > **Author's response addresses my comments**
> >
> > Thanks for your response. It addresses all my comments and I raise my score.

---

### Official Review · Reviewer_CscF · 2021-11-02

**Correctness:** 3
**Technical Novelty And Significance:** 2
**Empirical Novelty And Significance:** Not applicable
**Recommendation:** 8
**Confidence:** 3

**Main Review:**

Strengths:
1. From both theoretical and experimental perspectives, the authors demonstrate the effectiveness of combining randomized smoothing with ensembles in building certified robust classifiers.
2. Though the theoretical arguments include many assumptions, the authors carried out experiments to validate them.
3. They proposed Adaptive Sampling and K-consensus algorithms to reduce the computational cost, making their method more practical.

Weaknesses/Concerns:

While the empirical evidence are sufficient to support the effectiveness of using ensemble to boost the performance of RS, I have some concerns about the assumptions in the theoretical arguments.

It assumes that the "correlation between the logits of different classifiers has a similar structure but smaller magnitude than the correlation between logits of one classifier". Since different classifiers only differ in random seed for training, they should share behaviors in common. However, it is quite possible that such commonness is captured by $\mathbb{E}_l[f^l(x)]$, reflecting the underlying task, network architecture, dependence on $x$, etc., while the deviations of different models $\delta^i$ and $\delta^j$ are uncorrelated.

In fact, as shown in Figure 11(a) in the appendix, the off-diagonal blocks in the Delta subfigure are almost identical to the ones in the True covariance subfigure. It would be possible to reconstruct an argument about the variance reduction solely based on the assumption that $Cov(y^i_c, y^j_c)$ are near zero.

**Summary Of The Paper:**

In this paper, the authors propose using the aggregation of an ensemble of similar models as the base classifier in the randomized smoothing (RS). They show that the use of ensembles helps reduce the variance of the base classifier under noisy inputs, thus, improving the performance of RS. Both theoretical arguments and numerical experiments are included to support their idea. Further, the authors provide practical algorithms that significantly reduce the computational costs of their method.


**Summary Of The Review:**

The underlying topic of constructing a certified robust classifier is important. The main idea of the paper is sound and is supported by extensive experiment investigations. Further, the authors proposed algorithms to reduce the computational cost, making their method not only workable but also practical. There also includes empirical studies to validate (to some extent) the assumptions in their theoretical arguments. Hence, I would like to support the publication of the paper.

---

> ### Author Response · Authors · 2021-11-15
> **Response to Reviewer CscF**
>
> $\newcommand{Rt}{\textcolor{green}{CscF}}$
>
> We would like to thank reviewer $\Rt$ for providing valuable feedback and raising an interesting question, which we answer below.
>
> **It seems like the clean predictions $y_c$ of different classifiers are uncorrelated. Can you elaborate on how this is modeled and how the effect of the covariance of the clean component $y_c$ and the perturbation effect $y_p$ between classifiers vary?**
> This is an excellent observation. Indeed, we find that the predictions on unperturbed samples $y_c = f(x)$ are often effectively uncorrelated and capture this in our model with a very small correlation factor $\zeta_c=0.005$ for the example in Fig. 11.
> However, we must also consider the effect of perturbations and thus model the prediction $y=y_c+y_p$ as the sum of the clean component $y_c$ (identical across perturbations) and the perturbation effect $y_p$ (different for every perturbed sample). For the latter, we observe a non-zero covariance, which we capture in our model with $\zeta_p=0.85$ (shown at the bottom of Fig. 11). As the clean component does not change across perturbations, it has no impact on the variance of the distribution of predictions on perturbed samples but simply translates this distribution.
> We observe that the natural accuracy of our ensembles improves much less than its certified accuracies (see table below). This can only be explained if the variance of the perturbation effect $y_p$ decreases due to ensembling. Otherwise, individual constituting models would necessarily have to outperform the ensemble on every sample, as at least one of them would have a larger classification margin.
>
> The table shows the improvement in certified accuracy from 1 to 10 RN110 on CIFAR10 ($\sigma_\epsilon = 0.5$) at different radii $r$ and the natural accuracy.
>
> ||Nat|r = 0.00|r = 0.25|r = 0.50|r = 0.75|r = 1.00|r = 1.25|r = 1.50|r = 1.75|
> |---|---|---|---|---|---|---|---|---|---|
> |**Gaussian (abs %)**|2.2|3.2|6.2|7.6|7.6|7.8|5.0|4.2|3.0|
> |**Consistency (abs %)**|2.0|1.8|4.2|0.6|2.8|2.6|2.2|3.8|3.4|
> |**Gaussian (rel %)**|3.08|4.86|11.44|18.01|23.46|35.45|33.78|38.89|45.45|
> |**Consistency (rel %)**|3.08|2.85|7.66|1.23|6.67|7.22|7.38|16.96|20.70|
>
> We hope to have answered the reviewer's questions, would be happy to provide a more in-depth answer, and look forward to their response to this rebuttal.

---

### Official Review · Reviewer_ZAuf · 2021-11-02

**Correctness:** 3
**Technical Novelty And Significance:** 3
**Empirical Novelty And Significance:** 3
**Recommendation:** 6
**Confidence:** 4

**Main Review:**

This paper has several strengths.

- The problem that this work is tackling is both of interest and importance.
- The theoretical analysis presented in section 5 along with figure 2 potentially motivates the use of model ensemble.
- The Adaptive Sampling proposed in section 6 algorithm is practical.
- The experiments conducted in this work are extensive. They cover several frameworks for training smooth classifiers and several datasets. In most of the setups, model ensemble resulted in improvements in the certified accuracy.

However, there are several concerns that need to be addressed:

- The main claim about the time reduction that CertifyADP provides is based on an unfair comparison with Certify. The amount of information that both algorithms provide is not the same. CertifyADP tells whether an instance is certified with a pre-chosen radius $r$ or not, however it gives no hint about the actual radius that the instance is certified with, which is the output of Certify. I am not trying to lower the importance of CertifyADP since in many practical situations, it is useful to know whether an instance is certified with a given radius or not, but the comparison against Certify needs to be fair.

- The writing of the paper can be significantly improved (especially section 5).


Here are few comments/suggestions:

- typo: In page 4 third paragraph, "$y_p$ to be zero-mean".

-  While the analysis in section 5 are interesting, consider shrinking it to include more experiments in the main paper such as Table 15. It is interesting to see that the ensemble of models trained by different frameworks outperforms individual models.

- Consider switching the locations of figures 5 and 6 since figure 6 is mentioned in the text before figure 5.

- The definitions of SampleRF, TimeRF, and KCR are in the last page in the paper. However, they are important to understand table 3 and the "Computational Overhead Reduction" paragraph.

- Figures 2 and 3 need more elaborative discussion.

**Summary Of The Paper:**

This paper integrates model ensembles with randomized smoothing to improve the certified accuracy. The methodology is motivated theoretically by showing the effect of model ensemble on reducing the variance of smooth classifiers. Moreover, it proposes an adaptive sampling algorithm to reduce the computation required for certifying with randomized smoothing. Extensive experiments were conducted on CIFAR10 and ImageNet datasets.

**Summary Of The Review:**

This work has several merits as pointed out in the main review. The key concern is the unfair comparison between Certify and CertifyADP presented in this work. Moreover, the writing of this paper can be vastly enhanced for better exposure.

---

> ### Author Response · Authors · 2021-11-15
> **Response to Reviewer ZAuf**
>
> $\newcommand{Ro}{\textcolor{purple}{ZAuf}}$
> $\newcommand{Rth}{\textcolor{blue}{RVtN}}$
>
> We thank reviewer $\Ro$ for providing insightful feedback and raising interesting questions, which we answer below.
>
> **Is the comparison between Certify and CertifyADP fair despite Certify returning more information?**
> Our paper is very open about CertifyADP being only directly comparable to Certify in settings where a specific radius of interest is known in advance, and we highlight this fact multiple times. We believe, however, that the setting we consider is an important one as:
>
> * In most deterministic certification approaches, only one predetermined radius is considered. CertifyAdp allows, for the first time, a comparison of speed (often cited as a major drawback of randomized smoothing) in a comparable setting.
> * Many works building on Randomized Smoothing operate in the fixed radius setting such as [1, 2] and would benefit greatly from the speed-up provided by CertifyAdp.
>
> If the reviewer believes it to be necessary, we are happy to make this clear in the abstract as well.
>
> **Can you improve references from the text to figures and improve the presentation (esp. Section 5)?**
> Yes, we appreciate the reviewer's feedback and have incorporated all concrete suggestions, including the correction of the typo, the improved discussion of Figures 2 and 3, referencing Figure 5 before Figure 6, and the introduction of SampleRF, TimeRF, and KRC in the updated version of our paper. We hope that with these changes, the reviewer will find our paper as well-written as $\Rth$.
>
> **Can the analysis in Section 5 be shrunk to include more experiments in the main paper, such as Table 15?**
> We believe that reducing Section 5 would come at the cost of clarity of exposition and hence could not include Table 15 in the main paper due to space constraints. However, we have expanded its discussion in the evaluation.
>
> We believe to have incorporated the reviewer's feedback, hope to have answered all of the reviewer's questions, are happy to go into more detail regarding any of them and look forward to their response to this rebuttal.
>
> **References**
>
> [1] Fischer et al. 2020, “Certified Defense to Image Transformations via Randomized Smoothing”, NeurIPS20
>
> [2] Fischer et al. 2021, “Scalable Certified Segmentation via Randomized Smoothing”, ICML21

---

> > ### Comment · Reviewer_ZAuf · 2021-11-20
> > **Thank you for your response**
> >
> > I would like to thank the authors for their response. I still believe that it is necessary to emphasize that CertifyADP and Certify are not directly comparable to each other since they provide different amount of information. To that end, I suggest emphasizing this both in the abstract and in the experimental results so that readers are clear about the comparison.

---

> > > ### Author Response · Authors · 2021-11-21
> > > **Reply to Reviewer ZAuf**
> > >
> > > We thank the reviewer for their feedback and now explicitly emphasize the requirement of predetermined radii also in the abstract and the paragraphs on “Computational Overhead Reduction” and “Adaptive Sampling Ablation” in the experimental evaluation. Please see the updated version of the abstract in the pdf as the abstract on OpenReview can not be updated at the moment.

---

### Author Response · Authors · 2021-11-15
**General Response**

$\newcommand{Ro}{\textcolor{purple}{ZAuf}}$
$\newcommand{Rt}{\textcolor{green}{CscF}}$
$\newcommand{Rth}{\textcolor{blue}{RVtN}}$

We thank all reviewers for their insightful questions and valuable feedback. We are particularly encouraged that they consider the tackled problem both interesting and important ($\Ro$, $\Rth$), our solution theoretically (well) motivated ($\Ro$, $\Rt$), Adaptive Sampling practical ($\Ro$, $\Rt$), our experimental evaluation extensive ($\Ro$, $\Rth$), and our paper well-written and well-organized ($\Rth$).

Below, we briefly outline the updates in the revised submission based on the reviews. We address individual questions of reviewers in separate responses.

**Paper Updates:**
* [Section 2, New App. B] We now extensively review extensions of randomized smoothing, which are orthogonal to our approach but give readers a broader overview of the field ($\Rth$).
* [Section 5] We have explained Figures 2 and 3 in greater detail ($\Ro$).
* [Section 7] We improved the writing in the evaluation to reference figures in the correct order  ($\Ro$) and defined KCR, SampleRF, and TimeRF earlier ($\Ro$, $\Rth$).
* [Section 7] While we could not add Table 15 to the main paper due to space constraints, we have added a more detailed description of the results ($\Ro$).
* [App. H.4.1] We added a paragraph to section H.4.1 of the appendix where we describe and visualize the effects of $\alpha$ and $\beta$ on certification with CertifyAdp ($\Rth$).
* [New App. H.3.7] We conducted experiments showing that our approach is also effective for the certification of $\ell_0$, $\ell_1$, and $\ell_\infty$ (see new section H.3.7 in the appendix) ($\Rth$).

---

### Decision · Program_Chairs · 2022-01-20

**Decision:**

Accept (Spotlight)

**Comment:**

This paper integrates model ensembles with randomized smoothing to improve the certified accuracy. The methodology is motivated theoretically by showing the effect of model ensemble on reducing the variance of smooth classifiers. Moreover, it proposes an adaptive sampling algorithm to reduce the computation required for certifying with randomized smoothing. Extensive experiments were conducted on CIFAR10 and ImageNet datasets.

The strengths of the paper are as follows:
+ In terms of significance of the topic, the problem tackled in the paper is significant and highly relevant.
+ The motivation of using model ensemble is clearly illustrated via a figure and well justified with theoretical analysis.
+ Algorithmically, the paper proposes Adaptive Sampling and K-consensus algorithms to reduce the computational cost, making the method more practical.
+ Experimentally, the paper exhibits competitive results against several frameworks for training smooth classifiers and on several datasets.